# Large-scale analysis and computer modeling reveal hidden regularities behind variability of cell division patterns in *Arabidopsis thaliana* embryogenesis

**Elise Laruelle[1,2], Katia Belcram[1], Alain Trubuil[2]\*, Jean-Christophe Palauqui[1]\*, Philippe Andrey[1]\***

[1]Université Paris-Saclay, INRAE, AgroParisTech, Institut Jean-Pierre Bourgin, Versailles, France; [2]Université Paris-Saclay, INRAE, MaIAGE, Jouy-en-Josas, France

**Abstract** Noise plays a major role in cellular processes and in the development of tissues and organs. Several studies have examined the origin, the integration or the accommodation of noise in gene expression, cell growth and elaboration of organ shape. By contrast, much less is known about variability in cell division plane positioning, its origin and links with cell geometry, and its impact on tissue organization. Taking advantage of the first-stereotyped-then-variable division patterns in the embryo of the model plant *Arabidopsis thaliana*, we combined 3D imaging and quantitative cell shape and cell lineage analysis together with mathematical and computer modeling to perform a large-scale, systematic analysis of variability in division plane orientation. Our results reveal that, paradoxically, variability in cell division patterns of *Arabidopsis* embryos is accompanied by a progressive reduction of heterogeneity in cell shape topology. The paradox is solved by showing that variability operates within a reduced repertoire of possible division plane orientations that is related to cell geometry. We show that in several domains of the embryo, a recently proposed geometrical division rule recapitulates observed variable patterns, suggesting that variable patterns emerge from deterministic principles operating in a variable geometrical context. Our work highlights the importance of emerging patterns in the plant embryo under iterated division principles, but also reveal domains where deviations between rule predictions and experimental observations point to additional regulatory mechanisms.

**\*For correspondence:**
alain.trubuil@inrae.fr (AT);
jean-christophe.palauqui@inrae.fr (J-CP);
philippe.andrey@inrae.fr (PA)

## Editor's evaluation

This manuscript presents a new and interesting work exploring stochastic and deterministic aspects of embryonic cell division in plants. In particular, the power of the proposed approach lies in the quantitative analysis of 3D cell geometries that is combined with quantitative computer modelling.

## Introduction

In multicellular organisms, cell division is one of the major mechanisms that subtend the elaboration and maintenance of functional tissue organizations, as observed for example in animal epithelia (*Lemke and Nelson, 2021*). In plants, division is the primary determinant of relative cell positions because the cellular wall forbids cell displacements and intercalations (*Fowler and Quatrano, 1997*). Deciphering the principles that underlie the positioning and orientation of division plane is thus a central question to understand organ development and morphogenesis (*Gillies and Cabernard, 2011*). The possibility that universal primary physical principles operate in cleavage plane selection

has led to the formulation of several geometrical rules relating division plane positioning to mother cell shape (*Minc and Piel, 2012*), such as Errera's rule of plane area minimization for cells dividing symmetrically (i.e. producing daughters of approximately identical sizes, *Errera, 1888*). Although they are essentially phenomenological, such rules have proved useful as proxys to highlight generic cellular mechanisms that may be shared between cells with varying morphologies.

Stochastic fluctuations, or noise, play a major role in developmental systems (*Meyer and Roeder, 2014*; *Cortijo and Locke, 2020*). For example, at the molecular level, transcriptional noise has been recognized as a source of heterogeneity in cell fates (*Meyer et al., 2017*; at the cellular level, noise in growth rate has been suggested to contribute to the robustness in the development of organ size and shape *Hong et al., 2016*; at higher levels, it has been proposed that stochastic fluctuations could subtend plant proprioception up to the organ and organism scales *Moulia et al., 2021*). However, in contrast with variability and heterogeneity in cell and tissue growth, stochasticity in the positioning of the cell division plane has received much less attention. A noticeable exception is the seminal work of Besson and Dumais, who showed that in several two-dimensional plant systems with symmetric divisions, a stochastic formulation of Errera's rule accounted better for observed division patterns than its deterministic counterpart (*Besson and Dumais, 2011*). In addition, the impact on tissue organization of deterministic and variable division rules has been examined from a statistical point of view (*Gibson et al., 2006*; *Sahlin and Jönsson, 2010*; *Alim et al., 2012*; *Wyatt et al., 2015*) but the combinatorics of cell patterns (possible spatial arrangements of cells) that can result from variable cell divisions has not been examined with a cellular resolution. Overall, systematic analyses of variability in division plane positioning and of its relations to cell shape and tissue topological organization are currently lacking.

Here, we used the embryo in the model plant *Arabidopsis thaliana* to fill this gap, taking advantage of the variable cell division patterns observed in this system after initial rounds of completely stereotyped cell divisions (*Mansfield and Briarty, 1991*; *Capron et al., 2009*). We combined 3D image analysis, cell lineage reconstruction, and computer modeling to systematically dissect the spatio-temporal diversity of cell shapes and cell divisions and to challenge the existence of a possible geometrical rule linking cell geometry and division plane positioning. Paradoxically, our quantifications revealed that cell shapes resulting from variable cell divisions were evolving within a restrained repertoire of possibilities, highlighting the existence of hidden geometrical constraints behind the apparent variability of division patterns. We tracked the origin of these constraints back to the mother cell geometry and show that most of the observed patterns could be interpreted in light of a recently proposed division rule relating cell shape and plane positioning (*Moukhtar et al., 2019*). Our results reveal a unifying principle behind stereotyped and variable cell divisions in *Arabidopsis* early embryo, suggesting stochasticity is an emergent property of the evolution of cell shapes during the first generations of cell divisions. Cases where observed patterns deviate from the rule illustrate how our model can highlight domains where, beyond cell geometry, additional regulators may be involved in the positioning of the division plane.

## Results

In *Arabidopsis thaliana*, the fourth round of cell division leads to a 16-cell (16 C) embryo where four different domains can be distinguished based on their longitudinal (apical or basal) and radial (inner or outer) location (*Figure 1A*). The first four rounds of cell divisions follow invariant patterns, which can be predicted based on cell geometry (*Moukhtar et al., 2019*). Hence, 16C embryos exhibit cell shapes that are specific to each of the four domains (*Moukhtar et al., 2019*) and present invariant, symmetrical radial cell organizations in both the apical and the basal domains (*Figure 1BC*).

Here, we examined whether the stereotypical nature of cell shapes and patterns was maintained during late embryo development within each domain. We analyzed cell shapes and cell patterns over ~100 embryos between 1C and 256C stages (rounds 1–8 of cell divisions from the 1C stage). In accordance with previous observations (*Yoshida et al., 2014*), we initially observed that, from generation 5 onwards, the basal part of the embryo showed little variability in cell shapes and spatial arrangements, leading to a preserved radial symmetry across domains and individuals (*Figure 1DE*). On the contrary, shapes and arrangements of cells were highly variable in the apical domain. Different orientations and topologies of cell divisions were observed among the different quarters in a given individual as well as among different individuals (*Figure 1DE*). This variability resulted in a loss of radial symmetry of cell

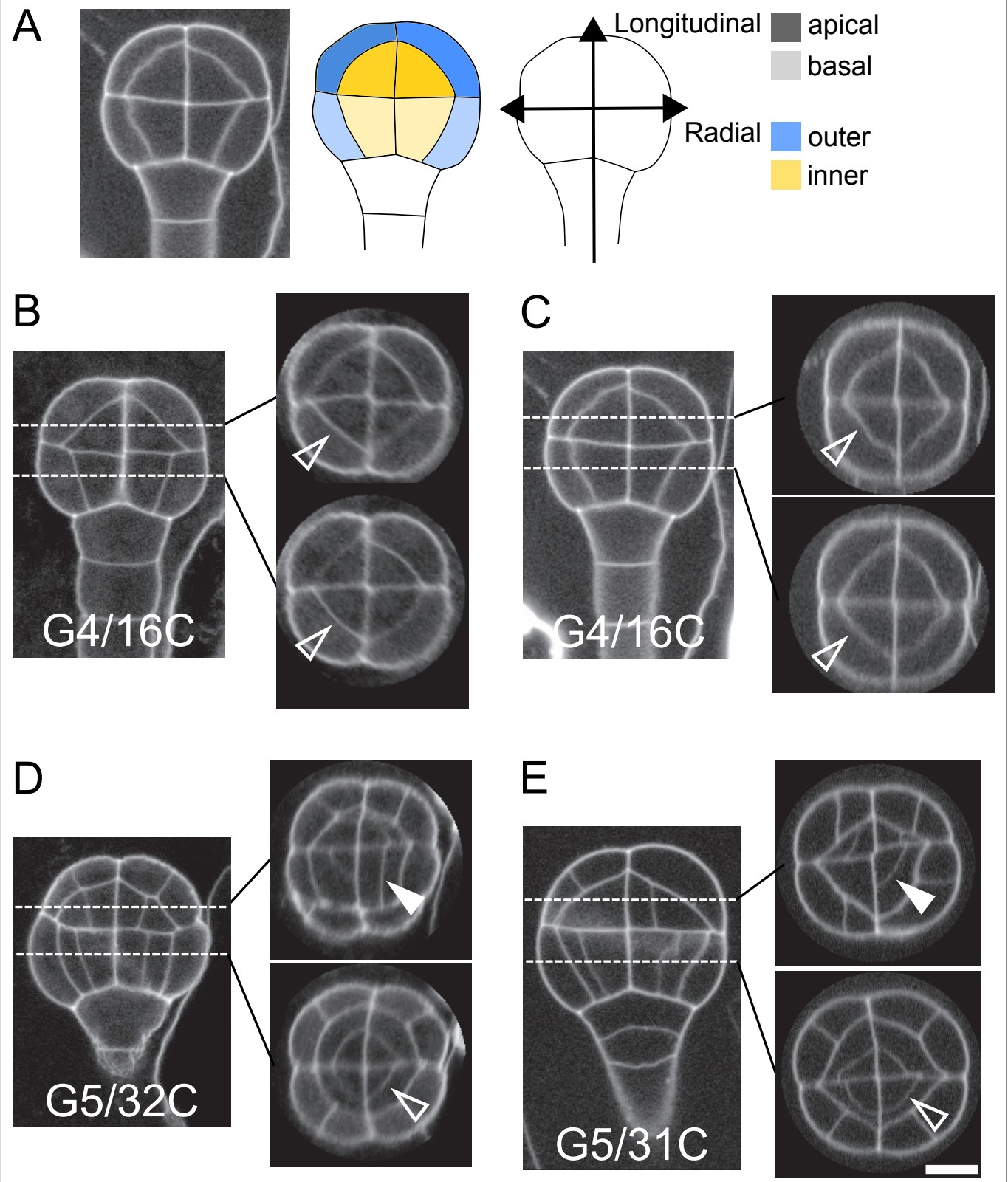

**Figure 1.** Variability within and between embryos in cell shapes and cell arrangements. (**A**) The four embryo domains defined by longitudinal and radial axes at stage 16C (longitudinal view): apical/basal × outer/inner. (BC) Invariant patterns in embryos up to generation 4 (16C). (DE) Starting from generation 5, embryos show variable cell shapes and cell patterns in the apical domains, both between individuals and between quarters in a given individual. Patterns in the basal domains show little or no variability. Some of the new interfaces at G4 (BC) and at G5 (DE) are labeled using arrow heads (*Empty*: invariant patterns; *Filled*: variable patterns). Scale bar: 10 μm.

organization in the apical domain (*Figure 1DE*). To better characterize and understand the origin of this variability, we conducted an in-depth quantitative analysis and modeling study of cell shapes and division patterns.

## Diversity in cell shape is domain-specific

To quantitatively describe cell patterns, we first focused on the diversity of cell shape in the embryo and in its four principal domains defined from the 16C stage (apical/basal × inner/outer). For each embryo, cells were segmented in 3D and their lineage reconstructed back to the 1C stage by recursively merging sister cells (*Figure 2A*). To this end, sister cells were identified and paired so as to minimize wall discontinuities in reconstructed mother cells (see Material and methods).

For a given mother cell, two features of the division plane determine the geometries of the daughter cells. First is the orientation of the division (e.g. anticlinal or periclinal), related to which mother cell walls are intersected by the division plane. Second is where the division plane is anchored on these walls and where it passes through the mother cell space. The orientation defines the shape (in the topological sense) of the daughter cells, that is, the morphological information that remains unchanged under position, rotation, scale or other linear and non-linear geometrical transformations such as anisotropic scaling, shearing, and bending. Plane positioning determines the lengths of cell edges, so that different cell geometries can be obtained for a same orientation of division. For example, cells in the inner apical domain at the 16C stage result from periclinal divisions of apical 8C cells and can all be described as tetrahedral pyramids (same shape) even though none of these cells have the same edge lengths (different geometries) (*Figure 1BC*). To classify cells into different shape categories resulting from varying orientations of cell divisions, one should thus rely on topological information only.

To this end, we introduced a new cell shape topology descriptor defined as the cumulative number of division planes that were positioned through generations to generate a given cell (*Figure 2B*). This number, referred to as the number of faces, was automatically computed from cell lineages reconstructed back to the initial 1C stage, which contains two faces (see Material and methods). A key advantage of this descriptor is to provide a robust, objective and unambiguous description of cell shape. Contrary to the number of neighbors or of geometrical facets, the number of division faces only depends on the topology of successive divisions that generated the considered cell, is independent of divisions in neighbor cells and is insensitive to geometrical fluctuations in the positioning of division planes and to their curvature.

We first applied this descriptor to analyze cell shapes up to the 16C stage (*Figure 2C*). The truncated sphere and half-sphere cell shapes of stages 1C and 2C have two and three-face shapes, respectively. The truncated sphere quarter at 4C has four division faces and is thus topologically equivalent to a tetrahedron. At stage 8C, the apical cells also have four faces but a new shape type is observed in the basal domain where cells have five faces, thus being topologically equivalent to a prism with a triangular basis. At stage 16C, a new cell shape with six faces was observed, being topologically equivalent to a cuboid. For each of the first four generations, each embryo domain (one domain from 1C to 4C, two domains at 8C, four at 16C) contained exactly one cell shape. These results are consistent with the stereotyped nature of cell division patterns until 16C stage. In addition, our analysis shows that at the whole embryo scale each generation corresponded to the introduction of a new cell shape with a unit increase in the number of division faces.

Over the next four generations (G5 to G8), we found that more than 99% embryonic cell shapes were distributed over the three main cell topologies already present at stage 16C, corresponding to shapes with four (3.6%), five (21.9%), and six (73.7%) faces (*Figure 2D*). From G4 onward, cell shapes progressively accumulated in the six-face (cuboid) shape category, which eventually represented more than 90% of the cells at G8 (*Figure 2E*). The systematic unit increase in the number of faces at each generation between G0 and G4 was no longer observed after G4. Hence, the transition between generations 4 and 5 (16C-32C) corresponded to a rupture in the dynamics of embryonic cell shapes. Cell shape heterogeneity, quantified by the entropy of the distribution of the number of cell faces at each generation, culminated at G4 and progressively decreased during the subsequent generations (*Figure 2E*).

The evolution of cell shapes at the whole embryo scale masked large differences among the four domains. Indeed, the domain-specific analysis of cell shapes showed that from generation 4 onward

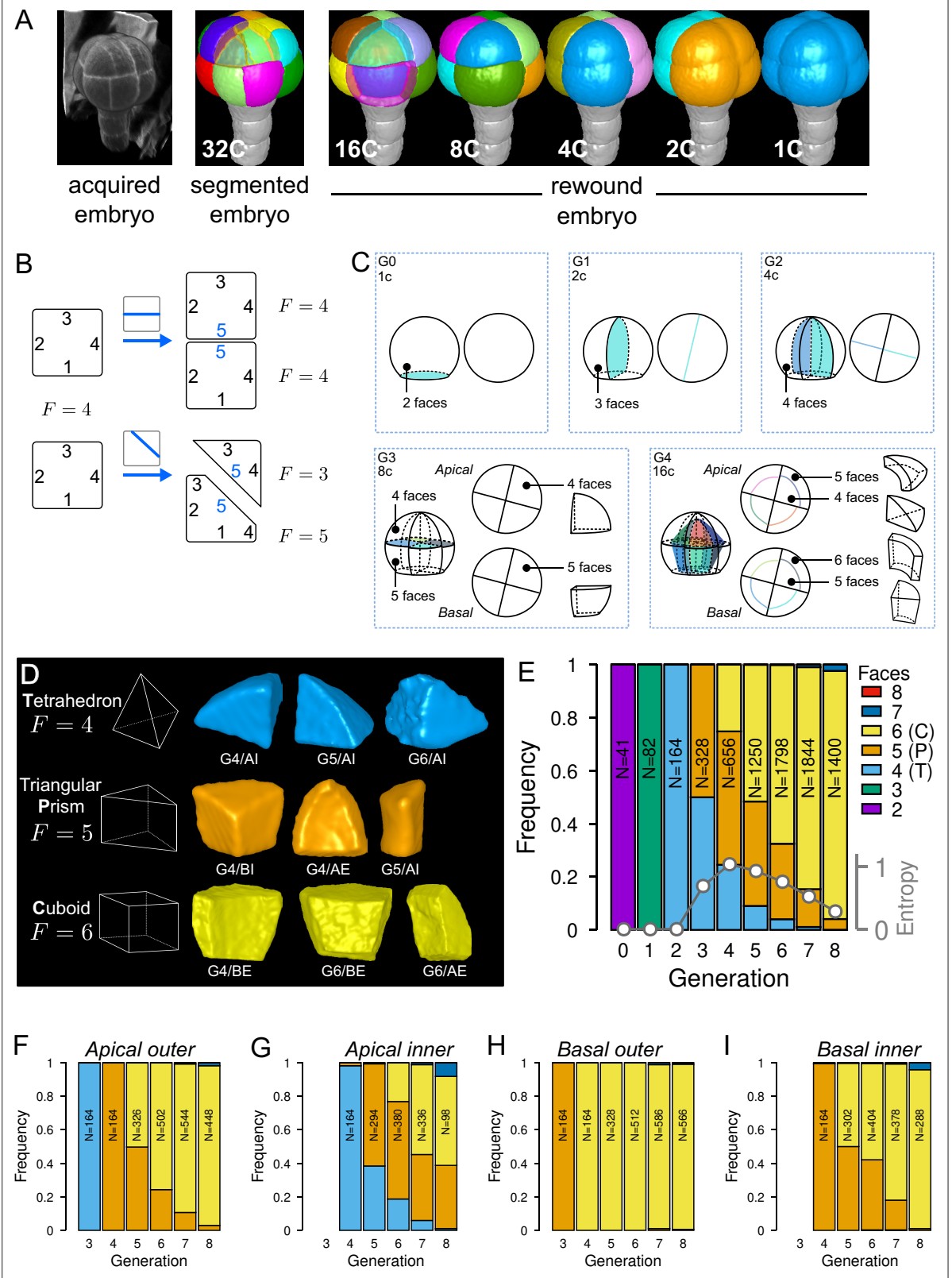

**Figure 2.** Cell shape diversity in *Arabidopsis thaliana* early embryogenesis. (**A**) Summary of 3D image analysis pipeline: 3D cell segmentation of confocal image stacks and cell lineage reconstruction by recursive merging of daughter cells. At 32C and 16C stages, some cells are shown transparent to visualize inner cells. (**B**) Classification of cell shapes based on the number of division interfaces. The scheme illustrates how the number of faces $F$ may change during a division. In both examples, a cell with initially four faces divides. The number of faces in daughter cells depends on the positioning

*Figure 2 continued*

of the new interface and of whether all original faces are represented in the daughter cells. (**C**) Shape classification during the first four generations. (**D**) Samples of the three main classes of cell shapes observed during the late four generations. (**E**) Proportions of shape classes over the whole embryo during the first eight generations. *C*: Cuboid; *P*: Prism; *T*: Tetrahedron. *Grey plot*: entropy of the distribution among the different shape classes. (**F–I**) Same as (**E**) over the outer apical (**F**), the inner apical (**G**), the outer basal (**H**), and the inner basal (**I**) domain. N: number of observed and reconstructed cells.

The online version of this article includes the following source data for figure 2:

**Source data 1.** Cell shape measurements.

there was almost no variability in the basal outer domain, where all cells remained in the six-face shape category (***Figure 2H***). The inner apical domain exhibited the largest variability in cell shape, with cells having four, five and six-faces observed through several consecutive generations (***Figure 2G***). In the basal inner and in the apical outer domains, the diversity was intermediate, with most cells distributed between the two categories of five and six-face shapes (***Figure 2F and I***). The dynamics were also similar in these two domains, with a continuously increasing proportion of six-face cells.

Overall, these results quantitatively confirmed the visual observations that cell patterns in the apical domain were more variable than in the basal domain. However, our analysis revealed at the same time a limited range of diversity in the topology of cell shapes, with most cell shapes falling within one out of three main categories. In addition, our data showed that the dynamics of shape changes during generations 5–8 differed from the dynamics observed during generations 1–4. Shape diversity increased until G4, before decreasing with an homogenization into the cuboid shape.

## Diversity in division patterns is domain-specific

Since cell shapes are determined by the positioning of division planes, we asked whether the diversity of cell shapes in the different domains could be related to domain-specific variability in the positioning and orientation of division planes. We examined this hypothesis by enumerating observed cell division patterns in each of the four embryo domains. Cell division patterns were characterized based on the shapes of the mother and of the daughter cells. In addition, we also took into account the relative orientation of the division planes within the embryo. For example, a triangular prismatic cell in the outer apical domain can divide according to three orientations into another prism and a cuboid (***Figure 3A***). These three possibilities were considered as distinct division patterns. Using lineage trees, we analyzed and quantified the frequencies of division patterns during the last four generations, using both observed patterns and patterns reconstructed at intermediate generations back to the 16C stage. Note that the absence of embryo bending at these stages ensured that the plane orientation in the embryo at the time of division could be correctly inferred even for patterns reconstructed from later stages.

Starting from the stereotyped cell patterns of 16C embryos, we found three major orientations of cell divisions in the outer apical domain at the G4-G5 transition (***Figure 3B*** and ***Figure 3—figure supplement 1***). Divisions in this domain were systematically anticlinal and oriented parallel to an existing cell edge, thus separating one vertex from the two other ones at the outer triangular surface of the cell. The transverse orientation (parallel to the boundary between the apical and basal domains) was less frequent than the two longitudinal orientations, suggesting a directional bias in the positioning of the division plane.

In the inner apical domain, we also found three main orientations of division planes, all oriented along the longitudinal axis of the embryo (***Figure 3D*** and ***Figure 3—figure supplement 2***). Only two of these orientations were parallel to an original vertical face of the cell. Divisions parallel to the horizontal face of the cells were extremely rare. As in the outer apical domain, these results suggested a preferential positioning of division planes along a limited number of directions.

In contrast with the apical domains, there was only one major orientation of division in each of the outer and inner basal domains (***Figure 3CE***). External cells systematically divided according to a longitudinal anticlinal division (intersecting their external face), with a division plane parallel to the lateral faces of the cell. Internal cells also divided longitudinally but along a periclinal division (parallel to their external face). This suggests even stronger constraints on the positioning of division planes within the basal domain compared with the apical domain.

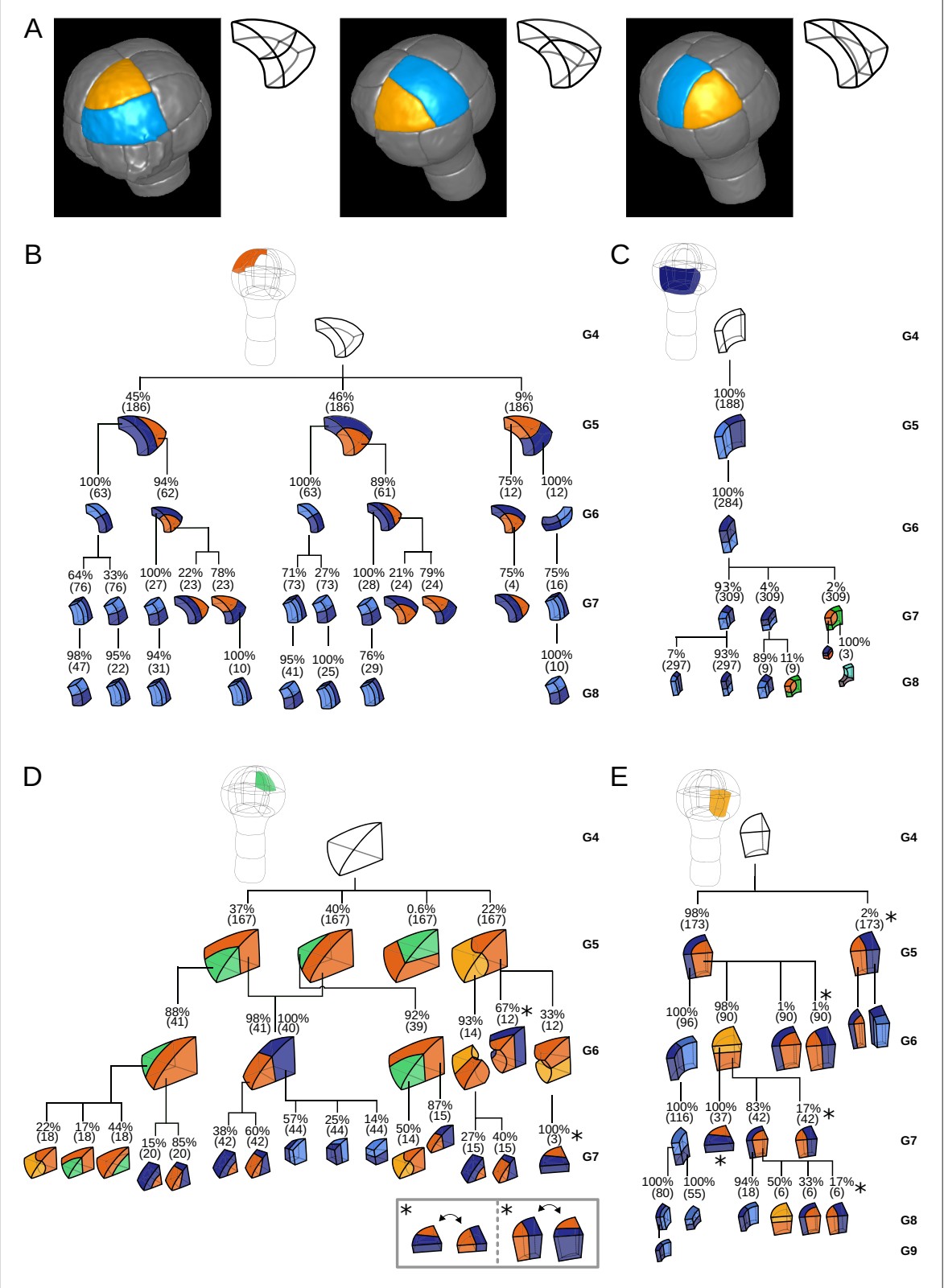

**Figure 3.** Reconstructed division patterns and cell lineages in the four embryo domains. (**A**) Classification of cell division patterns (illustration in the apical outer domain) based on mother and daughter cell shapes and on the absolute orientation of division planes within the embryo. (**BCDE**) Lineage trees in the apical outer (**B**), basal outer (**C**), apical inner (**D**) and basal inner (**E**) domains. Each tree shows the observed combinations of cell divisions as a function of cell shapes and of generations. At each generation, frequencies were computed based on both embryos observed at this generation and

*Figure 3 continued on next page*

*Figure 3 continued*

later embryos that had been reconstructed at this generation by recursively merging sister cells. Numbers in parentheses are the total numbers of cases over which the percentages were calculated. Exceptionally rare division patterns are omitted in (**B**) and (**D**) for the sake of clarity; complete versions are given in *Figure 3—figure supplement 1* and *Figure 3—figure supplement 2*. Asterisks correspond to symmetrical alternatives that were not distinguished in these trees.

The online version of this article includes the following figure supplement(s) for figure 3:

**Figure supplement 1.** Complete reconstructed cell lineages in the apical outer domain.

**Figure supplement 2.** Complete reconstructed cell lineages in the apical inner domain.

**Figure supplement 3.** Distance between cell division plane and cell center at different generations in *Arabidopsis thaliana* embryos.

**Figure supplement 4.** Volume-ratio of cell divisions at the G4-G5 transition in the four embryo domains (shown at G4 above the graph).

The contrast between the apical and the basal domains remained during subsequent generations, with strongly stereotyped division orientations in the basal domain, except for the division of the lower cells in the innermost domain at G6 (*Figure 3*). These results show that variability in the orientation of division planes was larger in the apical than in the basal domain during the latest four generations. By comparison with the stereotyped division patterns up to stage 16C, our analysis further corroborated that the transition between generations 4 and 5 corresponds to a rupture in the dynamics of division patterns.

## Division patterns correlate with cell shape topology

Since beyond stage 16C the embryo domains differ in the variability of both cell shapes and division patterns, we hypothesized that this variability could reflect shape-specific division patterns. We addressed this issue by exploiting reconstructed lineage trees to analyze division patterns in the three main cell shape categories that we identified.

Cuboid cells were found in all domains at several generations (*Figure 3B–E*). These cells almost exclusively divided into two cuboid daughter cells. Cuboid division resulting in a triangular prismatic daughter cell was only rarely observed. Hence, division of cuboid cells showed a strong auto-similarity, in that the mother cell shape was almost systematically preserved through the division. Another remarkable feature of the division of cuboids was spatio-temporal stationarity, since the division pattern of these cells was the same at all generations and in all four domains.

Cells with a triangular prism topology were also present in the four domains, when rare division patterns were also considered (*Figure 3B–E*). These cells showed two division patterns. The first pattern produced two triangular prisms as daughter cells, through a division parallel to the triangular faces. The second pattern yielded one triangular prism and one cuboid, through a division parallel to the quadrilateral faces. Hence, as for cuboid cells, cells with a triangular prism topology showed auto-similarity in their divisions patterns, even though they could also generate new cell shapes. In addition, they also showed spatio-temporal stationarity since their division patterns were similarly observed in all domains and generations where these cells were present.

Cells with a tetrahedral topology were only found in the inner apical domain (*Figure 3D*). They also exhibited two division patterns, one producing two triangular prisms and the other producing one triangular prism and one tetrahedron. Hence, auto-similarity in tetrahedral cells was not systematic. However, their division patterns were similar throughout successive generations, showing they were also exhibiting stationarity.

Together, these results show that each cell shape exhibited specific division patterns that were shared among different generations and among different locations within the embryo. The cuboid shape could be reached from any other cell shape according to the *tetrahedron→triangular prism→cuboid→cuboid* sequence. Hence, the cuboid shape represented an absorbing state because it could be reached from the two other shapes but tended to reproduce itself once reached. In contrast, the tetrahedral shape was the less stable state. These results explain the decreasing relative frequencies of the tetrahedral and triangular prism cell shapes through generations of cell divisions observed in the four domains (*Figure 2E*). Because of shape differences at stage 16C between the four domains, these results may also explain differences in variability of division patterns. For example, the large variability observed in the inner apical domain can be interpreted in light of the intermediate triangular prismatic shape between the tetrahedral and cuboid shapes. Inversely, the absence of shape variability in the outer

basal domain can be related to the absorbing state cuboid shape already present at G4 in this domain. However, shapes with identical topology were observed in domains with different variability levels in division orientations, as for example in the outer apical domain and in the inner basal domain that both have triangular prismatic cells at G4. Hence, other factors than cell shape topology alone are probably involved in the variability of cell division patterns.

## Graph theory of cell division reveals variability is constrained

To assess whether additional factors govern division patterns beyond cell shape topology, we asked whether observed division patterns matched predictions from topologically random divisions. To this end, we used graph theory to describe polyhedral cells and their divisions and to enumerate all possible combinations of dividing a cell based on its topology, disregarding the lengths of its edges. The three main cell shapes (tetrahedron, triangular prism, cuboid) observed during generations 5–8 are polyhedra composed of vertices (cell corners), of edges connecting vertices, and of faces delineated by edges. These shapes can all be represented as planar graphs and displayed using 2D Schlegel diagrams (*Grünbaum, 2003*). These representations are obtained by projecting the 3D cell shapes in a direction orthogonal to one of their faces (*Figure 4A*). We represented cell divisions as graph cuts on these polyhedral graphs. A graph cut consists in removing some edges in a graph so as to partition the original vertices in two disjoint subsets (*Greig et al., 1989*). Representing divisions as graph cuts implies that divisions avoid existing vertices and edges, in accordance with the avoidance of four-way junctions. Hence, by removing some edges in the mother cell graph, any cell division resulted in the partitioning of the $V$ vertices of the mother cell into two subsets of $p$ and $V - p$ vertices. The graphs of the two daughter cells were obtained by adding new vertices at edge cuts and by introducing new edges between the added vertices (*Figure 4B*; Supplementary Information).

For a given mother cell with $V$ vertices, we found that any division separating $p$ vertices (with $p \leq V/2$) from the $V - p$ other ones could be fully described based on $p$ and the number of mother cell edges that were inherited by the daughter cell inheriting the $p$ vertices (Supplementary Information). We further found that in case the inherited edges formed a cyclic graph, the number of faces in one of the two daughter cells was the same as in the mother cell and was at most this number in the other daughter cell. In the case of an acyclic graph, however, at least one daughter would necessarily gain one additional face as compared with the mother cell (Supplementary Information). This theoretical result explains in particular why the number of faces in at least one daughter cell necessarily increases when a tetrahedral cell divides, since the division of tetrahedral cells exclusively corresponds to the acyclic case. This theory shows why tetrahedral cells cannot be an absorbing state and why they represent an inevitable source of cell shape diversity through their divisions.

For each cell shape topology, we determined all possible combinations of graph cuts under complete randomness. This allowed us to compute the expected proportions of daughter cells falling within each cell shape category (Supplementary Information). The theoretical distributions we obtained were significantly different from the observed distributions (*Figure 4C*), thus showing observed division patterns were not compatible with the hypothesis of randomly selected positioning of division planes.

Overall, the predictions made using graph theory under unconstrained, random divisions strongly contrast with observed division patterns, where no or only marginal increases in the number of faces was observed during the last four generations. Our analysis thus shows that the observed division patterns are constrained within a limited range of possible combinations.

## Division planes obey cell geometry constraints

To understand the origin of the limited variability in cell division patterns, we asked whether cell geometry could be sufficient to account for the observed division planes. We previously showed that, during the first four generations, diverse division patterns (symmetrical as well as asymmetrical, anticlinal as well as periclinal) could be predicted by a single geometrical rule according to which planes obey area minimization conditioned on the passing through the cell center (*Moukhtar et al., 2019*). The small distance between division plane and cell center observed during the late four generations (*Figure 3—figure supplement 3*) suggested that this rule could also hold beyond the first four generations. To examine whether this was indeed the case in spite of diverse division orientations (*Figure 3*)

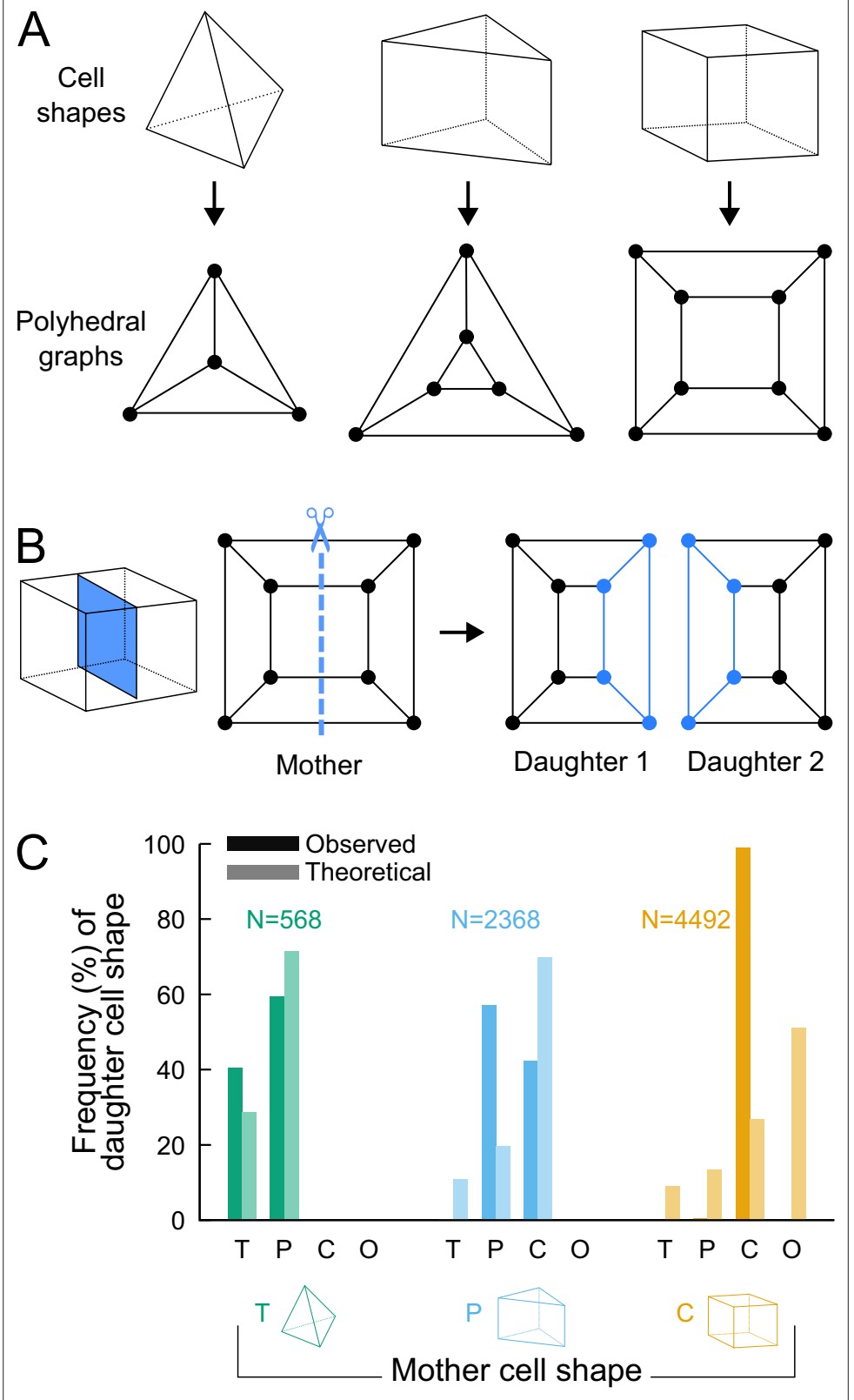

**Figure 4.** Analyzing cell divisions as graph cuts on polyhedral graphs. (**A**) The three main cell shapes and their corresponding polyhedral graphs shown as Schlegel diagrams. Dots in the graphs correspond to cell vertices and lines correspond to cell edges. (**B**) Cell division as graph cuts: illustration with the division of a cuboid shape. The division shown on the left corresponds to the removal of four edges in the mother cell graph (edges intersected

*Figure 4 continued on next page*

*Figure 4 continued*

by the dotted line). New vertices and edges added in the graphs of the two resulting daughter cells are shown in blue. (**C**) Observed and theoretical frequencies of daughter cell shapes during the division of each cell shape. Theoretical predictions were obtained under random graph cuts. *T*: tetrahedron; *P*: triangular prism; *C*: cuboid; *O*: others.

The online version of this article includes the following source data for figure 4:

**Source data 1.** Theoretical and observed frequencies of daughter cell shapes.

---

and volume-ratios (*Figure 3—figure supplement 4*), we compared observed division patterns at G5 to predictions derived from a computational model of cell divisions. We used a stochastic model that generated binary partitions of a mother cell at arbitrary volume-ratios, under the constraint of minimizing the interface area between the two daughter cells (*Moukhtar et al., 2019*) (see Material and Methods). This cell-autonomous model takes as input the cell geometry alone, ignoring the environment of the cell within the tissue. Several independent simulations with different volume-ratios were run for each reconstructed mother cell to sample the local minima of interface area in the space of possible binary partitions.

Running the model in synthetic shapes showed that repeating independent simulations at various volume-ratios generally produced several families of solutions (*Figure 5*). Each family corresponded to one of the possible combinations of graph-cuts in the polyhedral graph of the mother cell. The families could be visualized by plotting the distribution of simulation results based on surface area and distance to the cell center. For instance, simulations within a cuboid generated families corresponding to divisions parallel to two of the cuboid faces. In the distribution plots, such families appeared as vertically oriented clusters because of the similar areas but varying distance to the cuboid center (*Figure 5*). Other families corresponded to oblique divisions, isolating one vertex or one edge (*Figure 5*). These

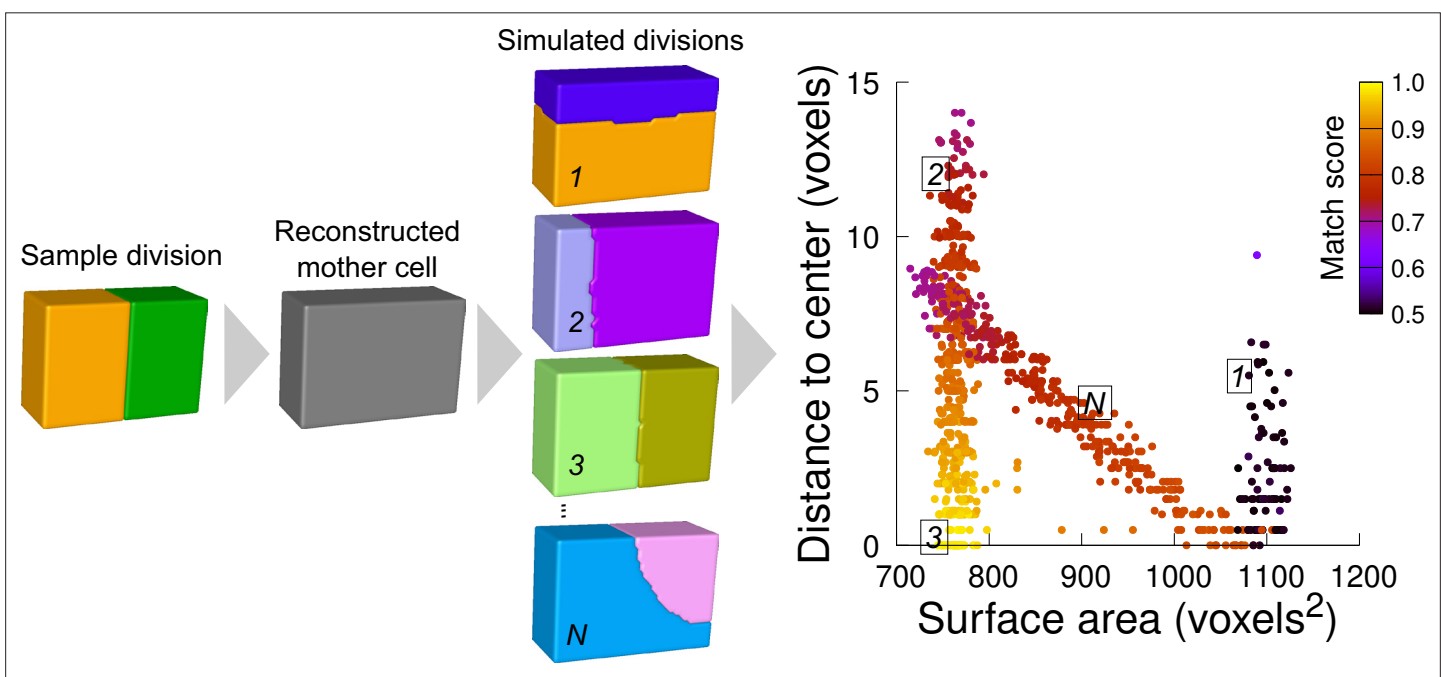

**Figure 5.** Computational strategy to analyze cell divisions: illustration with a synthetic example (symmetrical vertical division of a cuboid). Starting from a sample division, the mother cell is reconstructed and a large number of divisions at various volume-ratios is simulated. The distance from the cell center and the surface area of the simulated planes are computed. A match score, quantifying the correspondence with the sample division, is computed for each simulated division and represented in pseudo-color. In the present case, the graph shows several families of simulated planes. The location at the bottom left of the distribution of the simulations closest to the sample pattern shows that this division corresponds the minimum plane area among the solutions that pass through the cell center.

The online version of this article includes the following figure supplement(s) for figure 5:

**Figure supplement 1.** Quantifying the similarity between observed and simulated cell divisions.

families appeared as diagonally oriented clusters because area of these solutions increased when the distance to the center decreased.

We scored the similarity between simulation results and observed patterns based on a matching index. This index quantified how well a simulated pattern was reproducing the observed one based on the overlap between daughter cells in the two patterns (figure Supplement 1 and Material and Methods). This index ranged between 0.5 (minimal correspondence between simulation and observation) and 1.0 (perfect correspondence). For a sample division obeying the law of area minimization constrained by the passing through the cell center, the simulated divisions that match best the observed pattern should be located at the bottom left of the distribution plot (*Figure 5*).

We first examined divisions in cells of the outer basal domain, which obey a stereotyped symmetrical, anticlinal, and longitudinal positioning of the division planes (*Figure 3C* and *Figure 3—figure supplement 4*). For each cell, we ran 1000 independent simulations, which appeared sufficient to explore the space of area-minimizing partitions in a reproducible way (*Figure 6—figure supplement 1*). The distribution plots of simulated division planes based on surface area and on distance to the cell center were insensitive to potential segmentation errors (*Figure 6—figure supplement 3*) and were reminiscent of those observed in synthetic cuboid shapes (*Figure 6* and *Figure 6—figure supplement 4*; compare with *Figure 5*). Different clusters of simulated planes were observed, revealing the existence of several local minima of the interface area within the space of possible partitionings in these cells (*Figure 6A*). In spite of the variability in the geometry of analyzed cells (*Figure 6—figure supplement 5*), the simulated planes that matched the observed patterns were systematically found at the bottom left of the distribution plot (*Figure 6A* and *Figure 6—figure supplement 4*), showing that these matching planes were minimizing the surface area among the solutions that pass close to the cell center. Two other clusters of simulated planes, corresponding to either oblique or horizontal divisions, poorly matched observed patterns and had a larger interface area and/or a larger distance to the cell center. Hence, the anticlinal, highly symmetrical division of the basal outer cells at stage 16C of the embryo was perfectly predicted by the division rule. In most cells, the matching solutions were at the bottom of a cluster of solutions displaying a wide range of distances to cell center but comparable areas, corresponding to a family of parallel longitudinal divisions. This confirmed our previous result that, by the combined minimization of distance to cell center and of interface area, the rule can predict both the positioning of the division plane and the volume-ratio of the division (*Moukhtar et al., 2019*).

In the outer apical domain, where slightly asymmetrical, non stereotyped divisions were observed (*Figure 3B* and *Figure 3—figure supplement 4*), we ran the model in reconstructed mother cells that divided along the three main modes of division observed in this domain. As in the basal domain, the model generated different families of solutions within each mother cell (*Figure 6B–D*), showing the existence of different local minima of surface area for a given cell geometry. In each case, one of these cluster faithfully matched the observed pattern. The location of this cluster at the bottom left of the distribution plot suggested that for a given mother cell shape, the observed division plane could be predicted based on area minimization conditioned on the passing through the cell center (*Figure 6* and *Figure 6—figure supplement 6*, *Figure 6—figure supplement 7* and *Figure 6—figure supplement 8*). Remarkably, simulations belonging to the other, non-matching clusters, which were located farther from the bottom left of the distribution, corresponded to division patterns observed in other cells (*Figure 6B–D*). These data can be interpreted as showing the existence of three principal local minima of surface area in the space of partitionings of each apical outer cell. These minima are likely related to the order 3 rotation invariance of perfectly symmetric triangular prisms. Departure from perfect symmetry would turn one of these local minima into a global minimum that would be selected upon division. Our results also show that cells divide according to the area minimum that fits best with the same division rule that operates in the outer basal domain.

As in the outer basal domain, simulation results within basal inner cells (were observed divisions were stereotyped, periclinal and strongly asymmetrical; *Figure 3E* and *Figure 3—figure supplement 4*) were distributed among different patterns. However, a key difference with the outer domain was that a few, if any, simulations reproduced the observed divisions (*Figure 7A* and *Figure 7—figure supplement 1*). Since the probability of generating a given interface with the model is inversely related to its area, the absence or scarcity of reproduced observed patterns suggested that the periclinal divisions in the inner basal domain did not correspond to the global minimization of interface

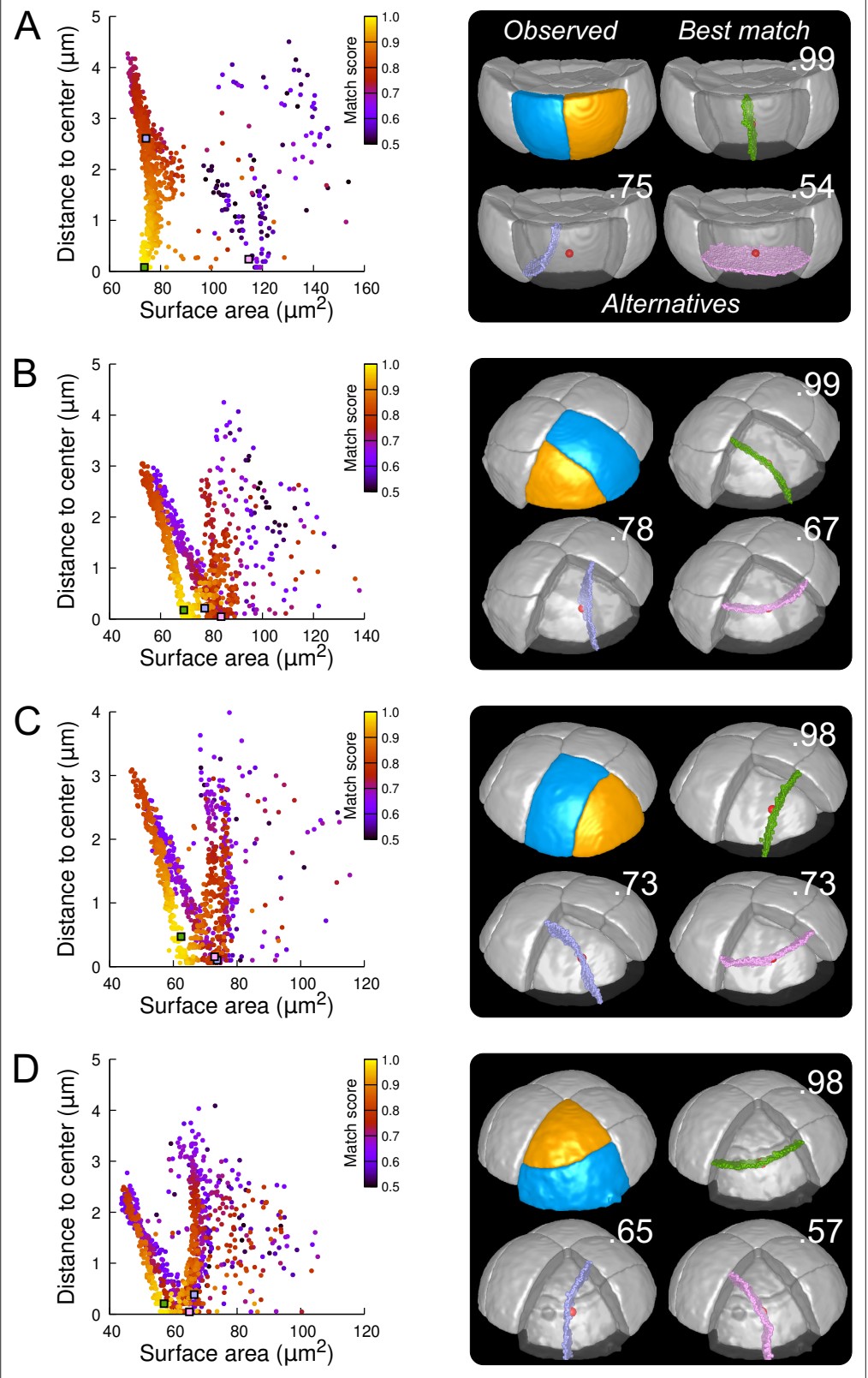

**Figure 6.** Modeling division patterns at G5 in outer cells based on geometrical features. (**A**) *Left*: distribution plot of simulation results in a basal outer cell (N=1000). Simulated planes are positioned based on their surface area and distance to the mother cell center. The dot color indicates the match score between simulated and observed planes. *Right*: observed daughter cells (*Blue* and *Orange*); three simulated planes are shown in the reconstructed

*Figure 6 continued on next page*

*Figure 6 continued*

mother cell (*Transparent*). *Green*: simulation matching best with observed pattern. *Lavender* and *Pink*: simulations with alternative orientations. Numbers show the corresponding match scores. *Red dot*: mother cell center. In the dot plot (*Left*), the positions of the three simulated planes are shown as squares with the same colors. (BCD) Same as (**A**) for three apical outer cells that divided along the three main orientations of division.

The online version of this article includes the following figure supplement(s) for figure 6:

**Figure supplement 1.** Convergence of the 3D computational model of cell division: cumulative variation of division plane area as a function of Monte Carlo cycle.

**Figure supplement 2.** Reproducibility of simulation results: distribution plots of two independent batches of 1000 simulation runs each.

**Figure supplement 3.** Robustness of simulation results to mother cell segmentation.

**Figure supplement 4.** Results of cell division simulations at the 16C-32C transition in basal outer cells: distance to cell center as a function of plane surface area.

**Figure supplement 5.** Mother cell shapes for simulations in basal outer cells.

**Figure supplement 6.** Results of cell division simulations at the 16C-32C transition in apical external cells dividing with a cuboid to the left: distance to cell center as a function of plane surface area.

**Figure supplement 7.** Results of cell division simulations at the 16C-32C transition in apical external cells dividing with a cuboid to the right: distance to cell center as a function of plane surface area.

**Figure supplement 8.** Results of cell division simulations at the 16C-32C transition in apical external cells dividing transversely: distance to cell center as a function of plane surface area.

area. This was confirmed by the fact that the rare simulations reproducing observed divisions had generally larger interface areas than alternatives passing as close to the cell center.

In the internal apical cells, where experimental variability was the largest (*Figure 3D* and *Figure 3—figure supplement 4*), we found different results depending on the orientation of the division. For cells where division occurred parallel to an existing interface (yielding a triangular prism and a tetrahedron as daughter cell shapes), we obtained results comparable to those obtained in outer apical cells. Several clusters of simulations were obtained within each cell, and the one reproducing the actual division was in most cases located at the bottom left of the distribution (*Figure 7B*; Figures *Figure 7—figure supplement 2* and *Figure 7—figure supplement 3*). In the other clusters, we observed simulated divisions that corresponded to patterns observed in other cells (*Figure 7B*). Hence, divisions in these cells were consistent with the existence of multiple local minima of interface area and with the selection, among these, of the minimum that also fits with the minimization of distance to the cell center. In cells dividing radially (yielding two triangular prisms for daughter cell shapes), some cells complied with this rule (*Figure 7C*; *Figure 7—figure supplement 4*) but we also found as many that did not. In the latter cells, several clusters corresponding to various division orientations were again observed. However, the cluster reproducing the observed division was either overlapping with other clusters or was located farther away from the heel of the distribution plot compared with the alternative clusters (*Figure 7D*). This showed that in these cells, the observed division was not unequivocally corresponding to the minimization of distance to the cell center and of interface area.

## Validation of model predictions

Simulation results obtained with our model suggested that asymmetries in mother cell geometry could bias the positioning of the division plane. We evaluated this prediction by examining the correlation between asymmetries in the mother cell geometry and the division plane orientation. We performed this analysis on the divisions of the 16C apical cells. For these cells, there was indeed, at the same time, strong self-similarity by rotation of the corresponding idealized shapes (tetrahedron in the inner part, triangular prism in the outer one) and large variability in the orientation of the division planes. For each reconstructed mother cell, we quantified its radial asymmetry by the ratio of left to right lengths and we quantified its longitudinal-to-radial asymmetry by the ratio of its longitudinal length to its maximal radial length (*Figure 8A*).

For internal apical cells dividing longitudinally with a triangular prismatic daughter cell on the left, the left radial length was on average smaller than the right one (*Figure 8B*, *Green*). The reverse was observed for the internal cells that divided with a triangular prismatic daughter located on the right

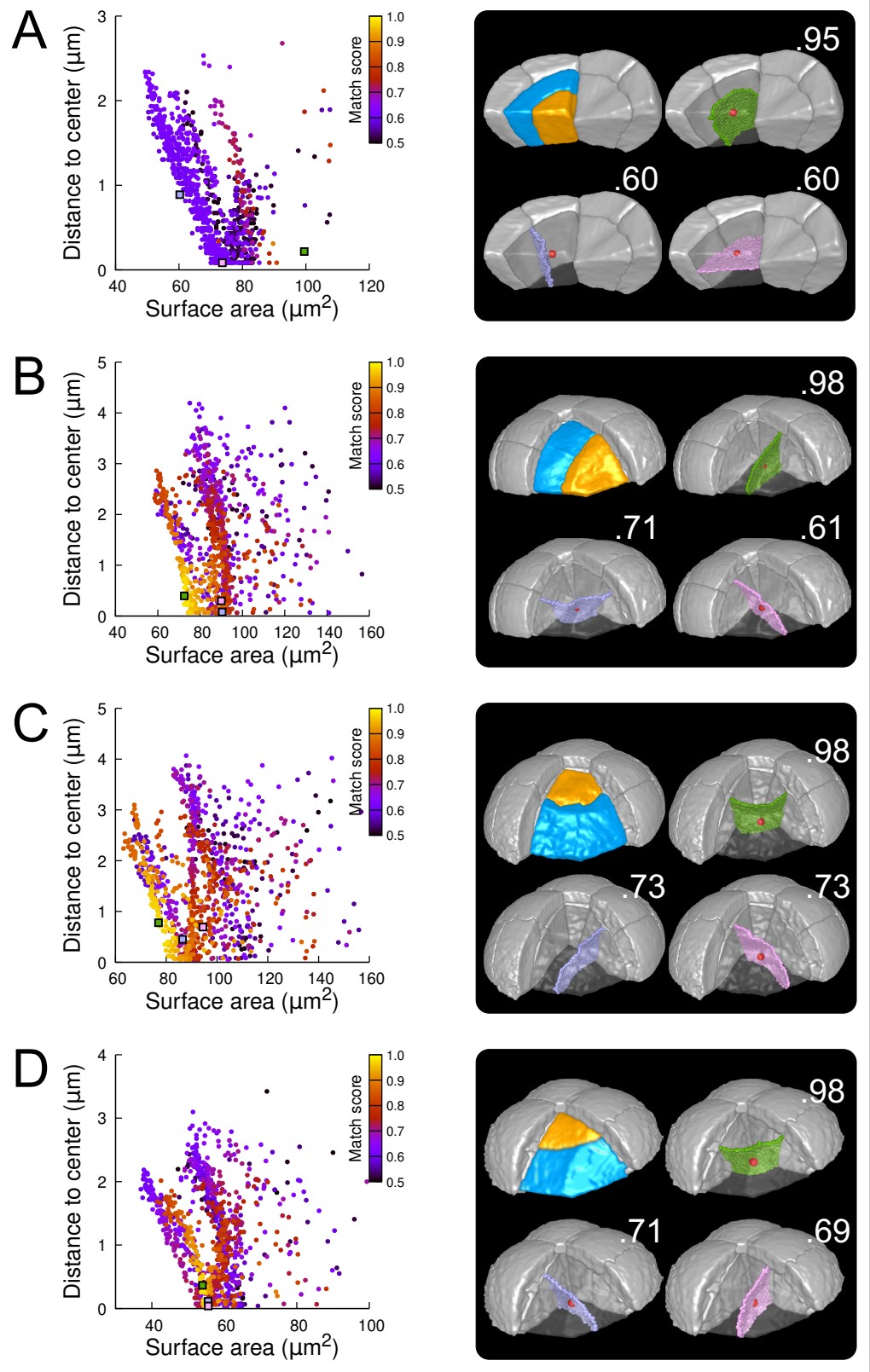

**Figure 7.** Modeling division patterns at G5 in internal cells based on geometrical features. (**A**) *Left*: distribution plot of simulation results in a basal inner cell (N=1000). Simulated planes are positioned based on their surface area and distance to the mother cell center. The dot color indicates the match score between simulated and observed planes. *Right*: observed daughter cells (*Blue* and *Orange*); three simulated planes are shown in the

*Figure 7 continued on next page*

*Figure 7 continued*

reconstructed mother cell (*Transparent*). *Green*: simulation matching best with observed pattern. *Lavender* and *Pink*: simulations with alternative orientations. Numbers show the corresponding match scores. *Red dot*: mother cell center. In the dot plot (*Left*), the positions of the three simulated planes are shown as squares with the same colors. (BCD) Same as (**A**) for three apical inner cells.

The online version of this article includes the following figure supplement(s) for figure 7:

**Figure supplement 1.** Results of cell division simulations at the 16C-32C transition in basal inner cells: distance to cell center as a function of plane surface area.

**Figure supplement 2.** Results of cell division simulations at the 16C-32C transition in apical inner cells dividing with a triangular prism to the left: distance to cell center as a function of plane surface area.

**Figure supplement 3.** Results of cell division simulations at the 16C-32C transition in apical inner cells dividing with a triangular prism to the right: distance to cell center as a function of plane surface area.

**Figure supplement 4.** Results of cell division simulations at the 16C-32C transition in apical inner cells dividing longitudinally and radially: distance to cell center as a function of plane surface area.

(*Figure 8B*, *Yellow*). For the internal cells that divided horizontally or longitudinally with no left/right asymmetry in plane positioning, there was no pronounced radial asymmetry (*Figure 8B*, *White* and *Pink*) but, compared with cells that divided longitudinally, they exhibited a larger longitudinal length (*Figure 8C*). Hence, in internal apical cells, the position of the division plane matched the geometrical asymmetry of the mother cell along different directions.

Similar trends were observed in the outer apical cells. Among these, cells dividing longitudinally with a cuboid daughter cell located on the left had on average a smaller left than right radial length (*Figure 8B*, *Turquoise*). The reverse was observed for the cells that divided with a cuboid daughter cell located on the right (*Figure 8*, *Orange*). As in the inner domain, the radial asymmetry was less pronounced for the outer apical cells that divided horizontally (*Figure 8B*, *Blue*). Compared with the inner domain, however, it was less clear whether their longitudinal length was larger than in cells dividing longitudinally (*Figure 8C*), which may be due to the limited number of cells that were observed to divide horizontally.

Overall, these results show that apical cells at 16C presented directional asymmetries and that division planes tended to be oriented parallel to the smallest cell length. This suggests that the diversity of division plane orientations for a given shape topology reflects geometrical diversity, in accordance with the predictions from our geometrical division rule.

## Attractor patterns buffer variability of cell division orientation

The above results show that from one generation to the next, there is large variability in cell division orientation in some embryo domains. Across several generations, the combinatorial possibilities between different orientations can potentially lead to a large number of distinct cell patterns. To determine whether this was indeed the case, we analyzed division patterns over two consecutive generations.

In the outer apical domain, three main orientations of cell divisions were observed at G5. Variability in division orientation was less pronounced in the subsequent generations, which presented alternation of division plane orientations (*Figure 3B*). As a result, similar cell patterns could be reached at G6 through different sequences of division events from G4. Over the 135 patterns observed at G6 in the apical outer domain, only 7 distinct sequences were observed. Two sequences were predominant and accounted for 40% (54/135; *Figure 9A*, *Top*) and 42% (58/135; *Figure 9A*, *Bottom*) of the observations.

In the protodermal layer of the basal domain, some variability was first observed at the transition between G6 and G7, where in 16 out of 303 cases (5.3%) the division plane was oriented transversely instead of longitudinally (*Figure 3C*). Similarly, some cells (19/297, 6.4%) at G7 divided longitudinally instead of transversely (*Figure 3C* and *Figure 9C*). Some cells in early heart stage embryos of our collection had already underwent an additional round of cell division, allowing to examine the evolution of such patterns. The cells that had exceptionally divided longitudinally at G7 led to daughters cells that divided transversely at the next generation, thus restoring at G9 the same 2×2 checkerboard

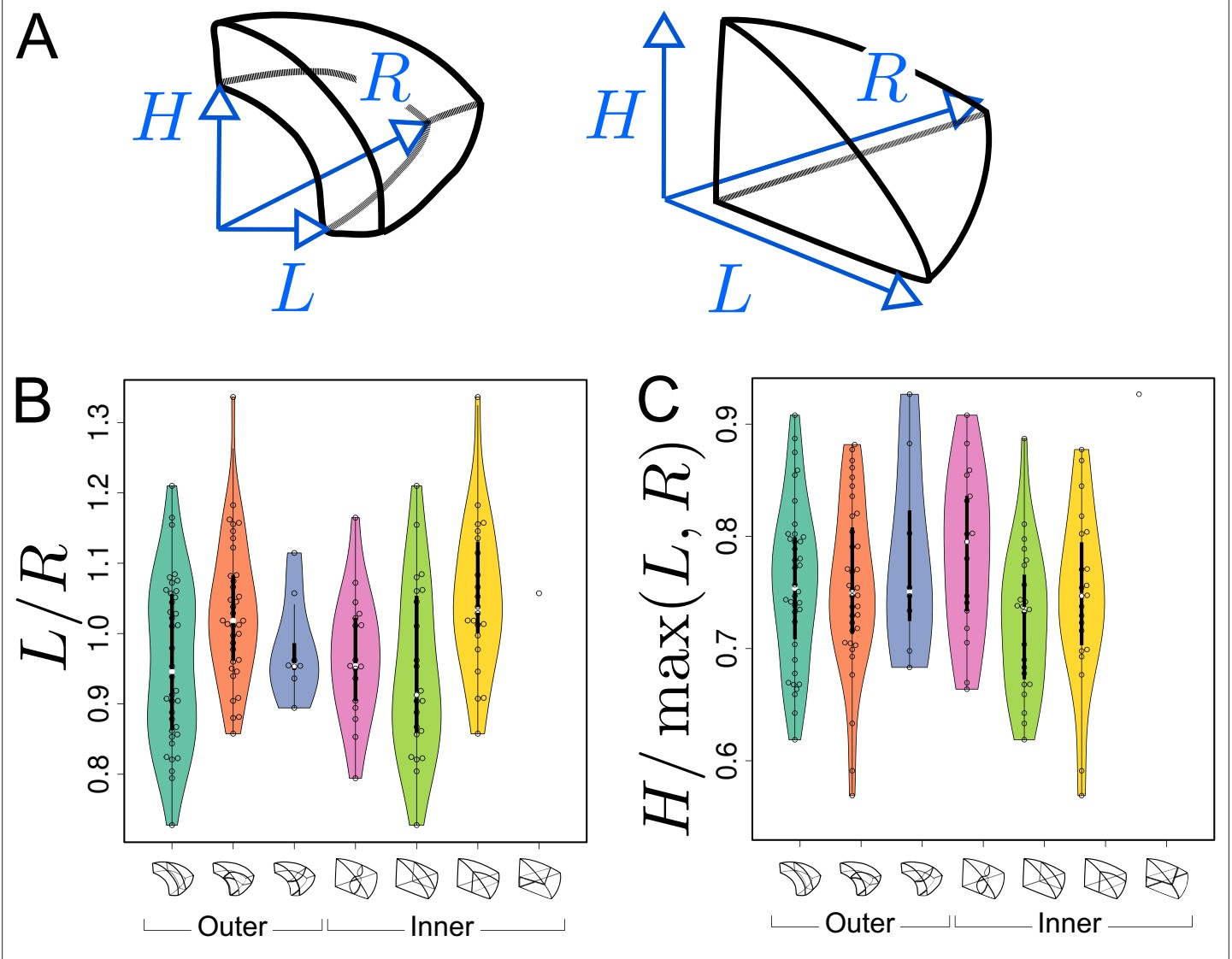

**Figure 8.** Asymmetries in mother cell geometry in the apical domain at stage 16C and their relations with division plane orientation. (**A**) Measured lengths of outer (*Left*) and inner (*Right*) mother cells. $H$: length along the longitudinal direction; $L$ and $R$: lengths along the left and right radial directions. (**B**) Radial asymmetry. (**C**) Relative longitudinal length. Measurements were performed on mother cells reconstructed at G4/16C from observed embryos at G5 or G6.

The online version of this article includes the following source data for figure 8:

**Source data 1.** Left/right length ratio measurements.

**Source data 2.** Longitudinal/radial length ratio measurements.

cell pattern than obtained along the transverse then longitudinal path followed in most embryos from G7 to G9 (***Figure 9C***).

In the inner basal domain, cell divisions were strongly stereotyped, following periclinal patterns that yield the precursors of the future vascular tissues (***Figure 3E*** and ***Figure 9B***, *Left*). At G5, however, 3 out of 153 cases (2%) in our dataset showed an anticlinal pattern (***Figure 3E*** and ***Figure 9B***, *Right*). One of these cases was reconstructed from an embryo acquired at G6. This allowed to observe that one of the two daughter cells of the anticlinal division at G5 had divided periclinally at G6, thus restoring the formation of a new cell layer as in the standard case (***Figure 9B***). This suggests that, in the inner basal domain also, similar cell patterns can be reached through different paths in spite of variability in division plane positioning.

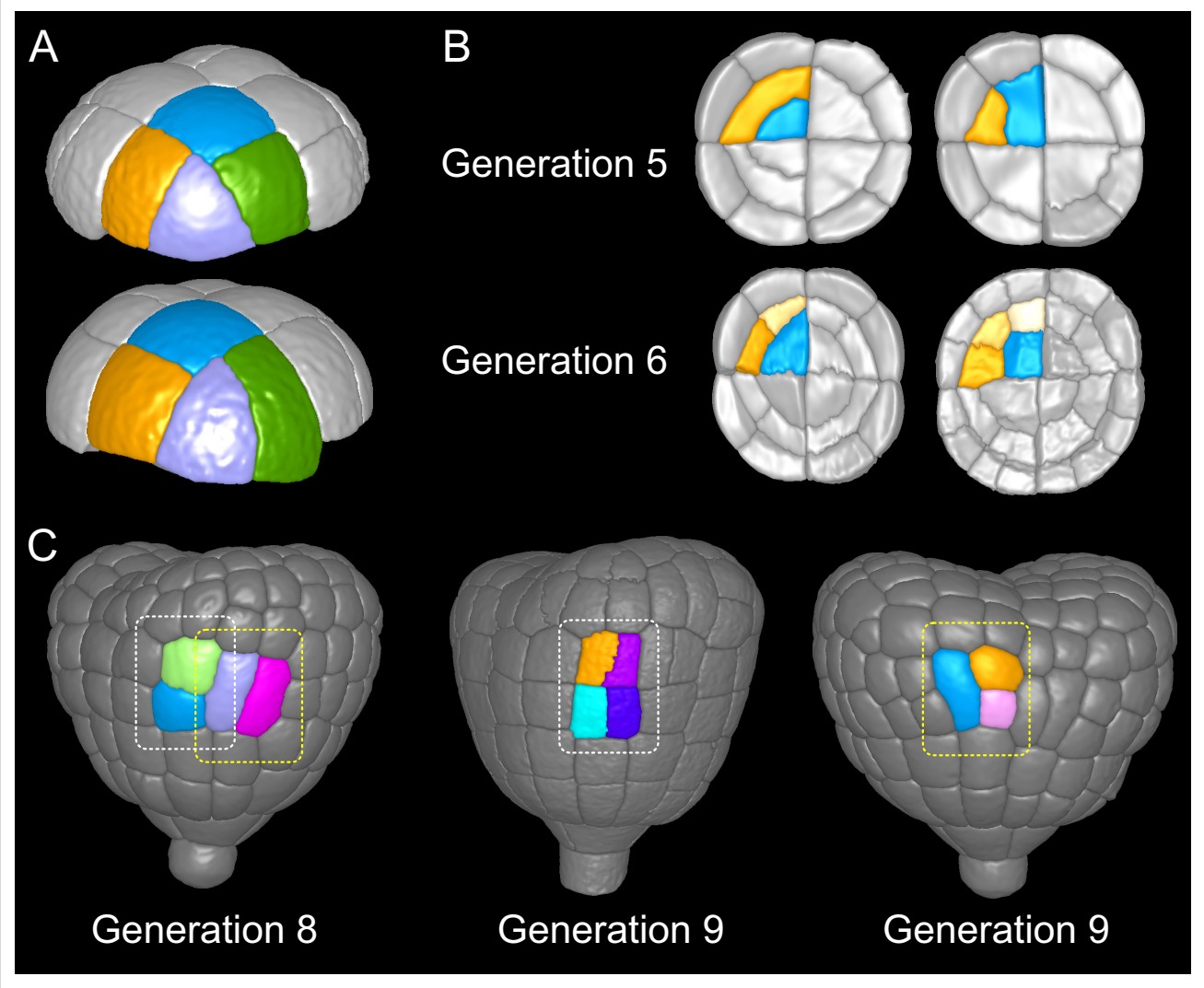

**Figure 9.** Attractor patterns buffer variability in division plane positioning. (**A**) Similar cell patterns observed at G6 in the apical outer domain that have been reached through distinct cell division paths from G4. (**B**) Main (*Left*) and rare (*Right*) division patterns in the inner basal domain at G5 and corresponding patterns at G6. (**C**) Main (*White box*) and rare (*Yellow*) patterns observed at G8 and G9 in the outer basal domain.

These results reveal the existence of attractor patterns, which are invariant cell arrangements that can be reached through different paths of successive cell divisions from stage 16C. The existence of attractor patterns suggests that a significant part of the variability in cell division orientation observed during the late four generations of embryogenesis is buffered when considering time scales that span several generations, thus ensuring the construction of robust cell organizations in spite of local spatio-temporal variability.

## Discussion

Previous attempts to decipher the principles that underlie the position and orientation of division planes have been focused on geometrical rules predicting the division plane position relatively to the mother cell geometry, and on their impact on global tissue organization and growth. Much less attention has been given to the prediction of tissue organization with a geometrical precision at the individual cell level. The *Arabidopsis* embryo is a remarkable model to address the existence and nature of geometrical division rules, as it presents invariant division patterns during the first four generations followed by intra- and inter-individual variable patterns for the next four generations. Here, we provided a detailed quantitative analysis of this variability and used theoretical and computational

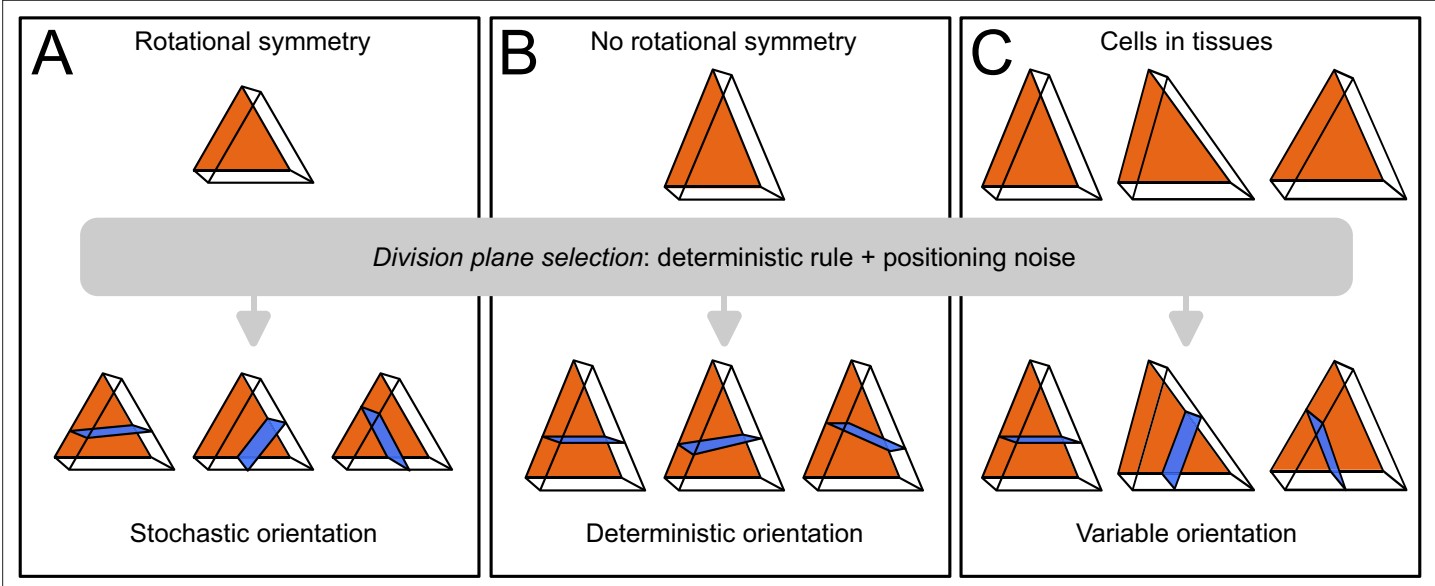

**Figure 10.** Schematic interpretation for the origin of variability in division patterns in *Arabidopsis thaliana* embryo. A deterministic selection of division plane orientation, combined with noise in the precise positioning of the division plane, can generate variable orientation patterns. (**A**) In rotationnally symmetric cells, different orientations are statistically equivalent, inducing stochasticity at the individual cell level; symmetry is lost in daughter cells due to positioning noise. (**B**) In non-symmetric cells, a single orientation is selected. (**C**) Observed cells present asymmetries differently aligned with respect to the embryo axes, resulting in variable division patterns at the tissue scale.

modeling of cell divisions to investigate its origin. We show that strong regularities are hidden behind the apparent variability and that most of the observed patterns can be explained by a deterministic division rule applied in a geometrical context affected by the stochasticity of the precise positioning of division plane.

Deterministic cell division patterns have been interpreted in light of geometrical rules linking cell shape to division plane (*Minc and Piel, 2012*). The shortest path rule, according to which cells divide symmetrically so as to minimize the interface area between daughter cells (*Errera, 1888*), has been shown to operate in several plant tissues such as fern protonema (*Cooke and Paolillo, 1980*), algae thallus (*Dupuy et al., 2010*), *Arabidopsis* meristem (*Sahlin and Jönsson, 2010*) or early embryo (*Yoshida et al., 2014*; *Moukhtar et al., 2019*). However, it was also shown that stochastic rules are required to account for division patterns in many tissues with 2D geometries (*Besson and Dumais, 2011*), as in some animal systems (*Théry et al., 2007*; *Minc et al., 2011*). Hence, a stochastic rule for division plane orientation would a priori be the most likely candidate interpretation of the variable division patterns we reported here in late *Arabidopsis* embryo. Our results point to a different interpretation for this variability. Indeed, for a given cell geometry, the observed plane orientation and position matched in most cases the global optimum according to the rule of area minimization conditioned on the passing through the cell center, and we could correlate the plane orientation with asymmetries in directional cell lengths.

Based on these results, we propose that variability in cell division patterns could originate from fluctuations in mother cell geometry rather than from the division rule. The tetrahedral and triangular prismatic shape topologies of apical cells at stage 16C are rotationally symmetric (they can superimpose to themselves after rotation). If cell geometries were perfectly symmetric, the various plane orientations (4 in inner apical cell, 3 in outer apical cells) would be equally probable according to the geometrical rule (*Figure 10A*). We hypothesize that actual geometrical deviations from perfect symmetry suppress this equi-probability and induce a single global optimum of plane orientation, which would be selected during the division (*Figure 10B*). Accordingly, variability in division plane orientations in the apical domain would not ensue from a stochastic division rule, but rather from a deterministic principle expressed within a varying mother cell geometry (*Figure 10C*). Note, however, that our sample sizes do not allow to definitively rule out a possible stochastic selection of division plane orientation, in particular in light of the results obtained in the inner apical domain with cells

dividing longitudinally. Further studies will be required to definitively distinguish between these two hypotheses and to further dissect the respective contributions of intrinsic (variability of plane positioning for a given cell geometry) and of extrinsic (variability due to fluctuations in mother cell geometry) noise in the selection of the division plane.

Our data reveal an abrupt change in the dynamics of cell shapes and cell division patterns at the transition between generations 4 (16C) and 5 (32C). Up to generation 4, division patterns were stereotyped and each generation corresponded to the introduction of a new cell shape with a unit increase in the number of cell faces. In contrast, we observed from generation 5 onward a strong variability in division patterns with a concomitant reduction in the variability of cell shape topology, as cell shapes progressively converged towards a single 6-face shape topology. Graph cut theory on polyhedral graphs together with our hypothesis of a deterministic division principle operating in a stochastic cell geometry offer a parsimonious interpretation of this apparent paradox. On the one hand, our theory shows that the division of the tetrahedral cells from generation 2 inevitably generates novelty with one obligatory prismatic daughter cell shape. We also show that triangular prismatic shapes that appear at generation 3 are theoretically twice less self-reproducible than the cuboid shapes that appear for the first time at generation 4. Beyond this stage, cell division through the cell center and area minimization tend to preserve the cuboid shape of the mother cell in the two resulting daughters. On the other hand, variability in division patterns emerges at generation 5 because of the almost, but not exactly, rotation-symmetrical cell geometries reached for the first time at stage 16C. Hence, our study reveals that a common underlying geometrical rule can account for cell division patterns with radically different traits, stressing the importance of geometrical feedback between cell geometry and division plane positioning in the self-organization of tissue architectures in *Arabidopsis* embryo. A parsimonious cellular machinery may be beneficial to ensure robustness in the building of complex cellular patterns.

Our interpretation of the variability in division orientations raises the issue of the origin of variability in cell geometry within a given cell shape category. In spite of genetic controls, any given division pattern is subject to random fluctuations affecting the precise positioning of the cleavage planes (*Schaefer et al., 2017*), even in strongly stereotyped systems (*Guignard et al., 2020*). Hence, rotational symmetry, if it were present at some stage, could not be preserved through cell divisions (*Figure 10A*). This noise in the positioning of division planes accumulate through embryo generations, resulting in non-perfectly symmetrical shapes at stage 16C. A modeling study previously reported the importance of stochastic positioning of cleavage planes at the 2C-4C transition in the patterning of vascular tissues (*De Rybel et al., 2014*). In our case, it is likely that errors accumulated over the 2C-4C and 4C-8C transitions contribute to the geometrical asymmetries that bias division plane positioning at the 16C-32C stage. Hence, our results strongly suggest that not only genetic patterning (*De Rybel et al., 2014*) but also division patterning could be influenced by the geometric memory of past stochastic events.

Several studies have highlighted the importance of noise and stochastic processes in plant developmental programs (*Korn, 1969*; *Meyer and Roeder, 2014*; *Hong et al., 2018*). At the cellular level, these processes have been described essentially for cell growth. For example, heterogeneity in cell growth patterns was shown essential for the robustness of organ shapes (*Hong et al., 2016*). Homeostatic mechanisms compensating for cell growth variability have been described. For example, at the cellular level, larger relative growth rates in smaller cells (*Willis et al., 2016*) or DNA-dependent dosage of a cell cycle inhibitor *D'Ario et al., 2021* have been proposed to subtend cell size homeostasis in the shoot apical meristem; at the tissue level, mechanical feedbacks have been described that buffer growth heterogeneities between cells (*Hervieux et al., 2017*). We reveal here in several embryo domains the existence of attractors in embryo cell patterns that can be reached through different division sequences, thus generalizing past observations in the root embryonic axis (*Scheres et al., 1995*). As for cell growth patterns, these attractor patterns can be interpreted as buffering heterogeneity in division plane orientation. Hence, our results reveal a new compensation mechanism at the cellular level that, in addition to known cell growth regulations, could operate in developing plant tissues to generate robust supra-cellular patterns.

Our quantifications showed a much larger variability of divisions patterns and cell shapes in the apical domain compared with the basal one. This difference can be related to the different cell shapes in the two domains at stage 16C. Tetrahedral shapes, which are an obligatory source of shape

variability through their division, are only present in the apical domain. Conversely, cuboid shapes, which represent an absorbing state, are only present in the basal domain. Different cell environments, with the basal cells constrained between the apical cells and the suspensor, may also contribute to less variability in the basal domain. Although the functional significance of this apical-basal contrast remains to be elucidated, one can hypothesize it could contribute to establish a specific tissue organization or mechanical context required for proper embryo growth and transition from a globular to a heart shape. Recent reports in both plants and animals emphasized the importance of the spatial organization of cell interfaces for tissue mechanical properties or cell-fate acquisition (*Guignard et al., 2020*; *Majda et al., 2022*).

Previous studies have modeled the topology of divisions in 2D. It was shown for example how an average of 6 neighbors per cell could emerge from random symmetrical divisions (*Graustein, 1931*; *Gibson et al., 2006*). Based on Markov chain modeling, it was also shown how steady-state distributions in the number of faces or of neighbors could be computed in proliferating epithelia (*Gibson et al., 2006*; *Cowan and Morris, 1988*). The topology of a 2D division in a polygonal shape can simply be modeled as a combinatorial choice of two polygonal edges (*Cowan and Morris, 1988*; *Gibson et al., 2006*). Unfortunately, this approach cannot be generalized to polyhedral cells in three dimensions. Here, we proposed a solution to this problem by modeling the topology of division in polyhedral cells as cuts on polyhedral graphs. The large differences between predicted daughter shape distributions under topologically random divisions of mother cells and observed distributions revealed the existence of strong constraints on division plane positioning at the 16C-32C transition. Though this is probably challenging, it would be of further interest to explore the potential of the proposed graph theoretical approach to address the existence of, and to theoretically derive, the asymptotic distributions of 3D shapes under random or more elaborate topological rules, as was done in 2D tissues (*Cowan and Morris, 1988*; *Gibson et al., 2006*).

The results of the present study show that the same geometrical rule that accounted for cell division patterns during the first four generations is also consistent with the positioning of division planes beyond the dermatogen stage. However, we found contrasting results among different embryo domains and, to a lesser extent, among different orientations of division. In the protodermal domains of both the upper and the lower domains, both the volume-ratios and the positioning of the cleavage interface could be accurately predicted following the geometrical rule. In contrast, divisions markedly departed from the rule in the lower inner domain. An intermediate situation was observed in the inner apical domain, where the rule accounted for all but the longitudinal radial orientation. Post-division changes in cell geometry can potentially alter predictions of plane positioning in mother cells reconstructed by merging their daughters, although such changes are probably moderate given the relatively limited cell growth at the stages we considered (*Yoshida et al., 2014*). Auxin signaling has been suggested as required for cells to escape the default regime of division plane minimization and to control periclinal divisions at the previous (8C-16C) generation of cell divisions (*Yoshida et al., 2014*), which could involve a modulation of cell geometry by auxin signaling (*Vaddepalli et al., 2021*). At subsequent generations, it has instead been reported that the first vascular and ground tissue cells divided periclinally along their maximal (longitudinal) length when the auxin response was impaired by a ARF5/MP mutation or local ARF inhibition (*Möller et al., 2017*). In the shoot apical meristem, cells preferentially divide longitudinally at the boundaries of emerging organs, where auxin responses are low (*Louveaux et al., 2016*). Hence, it is unclear whether specific auxin responses are involved in the longitudinal divisions observed in the inner domains. Mechanical forces have been shown to alter division plane orientations in in vitro-grown cells (*Lintilhac and Vesecky, 1984*), and it was shown in the shoot apical meristem that tissue mechanical stress could override cell geometry in the specification of plane positioning (*Louveaux et al., 2016*). It was also recently found that the orientation of cell division during lateral root initiation correlated with cellular growth (*Schütz et al., 2021*). Hence, one can speculate that the differences in cell environments between the inner and the outer embryo domains may induce different mechanical contexts with differential impacts on the determination of the division plane orientation.

## Materials and methods

### Sample preparation and image acquisition

#### mPS-PI staining

*Arabidopsis* siliques were opened and fixed in 50% methanol and 10% acetic acid five days at 4 °C. Samples were rehydrated (ethanol 50%, 30%, 10% and water) then transferred 3 hours in a 0.1 N NaOH 1% SDS solution at room temperature. Next, samples were incubated 1 hr in 0.2 mg/ml $\alpha$-amylase (Sigma A4551) at 37 °C and bleached in 1.25% active Cl_30-60 s. Samples were incubated in 1% periodic acid at room temperature for 30 min and colored by Schiff reagent with propidium iodide (100 mM sodium metabisulphite and 0.15 N HCl; propidium iodide to a final concentration of 100 mg/mL was freshly added) overnight and cleared in a chloral hydrate solution (4 g chloral hydrate, 1 mL glycerol, and 2 mL water) few hours. Finally, samples were mounted between slide and cover slip in Hoyer's solution (30 g gum arabic, 200 g chloral hydrate, 20 g glycerol, and 50 mL water) using spacers.

#### Confocal microscopy and image acquisition

Acquisitions were done with a Zeiss LSM 710 confocal microscope as described previously (***Truernit et al., 2008***). Fluorescence signals were recorded using a 40 x objective and digitized as 8-bit 3D image stacks with a near-to-optimal voxel size of 0.17×0.17×0.35 μm³.

### Image processing and analysis

Noise in acquired 3D images was attenuated by applying Gaussian smoothing (with parameter $\sigma = 0.5$) under the Fiji software (***Schindelin et al., 2012***). Cells were segmented by applying the 3D watershed transform (***Vincent and Soille, 1991***) to images after non-significant minima had been removed using minima imposition (***Soille, 2004***). The two operations were performed using the Morphological Segmentation tool of the MorphoLibJ suite (***Legland et al., 2016***). All segmentations were visually checked and a modified version of the MorphoLibJ plugin was developed to correct over- and under-segmentation errors, if any, based on the interactive modification of watershed initialization seeds.

The cell lineages were manually back tracked, processing embryos from the younger to the older ones (using the number of cells as a proxy for developmental stage). Based on the cellular geometries and organizations, sister cells were paired so as to minimize wall discontinuities in reconstructed mother cells. Ambiguities, as observed for example at the 2C-4C transition or later in the outer basal domain where four-way junctions could be observed at the external surface of the embryo, were resolved by examining cell interfaces in 3D. Indeed, actual four-way junctions were rare in 3D, as penetrating inside the tissue generally revealed a transition from a cross pattern formed by the four cells to a double-T pattern, thus revealing a former division plane that had been reached on its opposite sides by two more recent ones. Over a total of 4285 division patterns, we observed 12 cases (0.3%) at the 2C-4C generation of divisions where ambiguity could not be resolved this way. This had no impact on our analyses because of the symmetry between the two possible interpretations at this stage. We observed 16 cases (0.4%), all but one in the outer domains, in advanced stages. We resolved these ambiguous cases by assigning them to the most frequent configuration over the whole dataset of embryos. Lineage reconstruction was performed using TreeJ, an in-house developed Fiji plugin. Reconstructed cell lineage trees were exported as Ascii text files for further quantitative analysis.

Segmented images and lineages trees were processed under Matlab (***MATLAB, 2012***) to localize cells within the embryo and to assign them to embryo domains (inner or outer, apical or basal). Cell volumes were obtained by multiplying the number of voxels of each cell by unit voxel volume (product of spatial calibration in XYZ directions). Mother cells were reconstructed by merging the segmentation masks of daughter cells. The mother cell center was computed as the average voxel position in the mother cell mask. For each division, the volume-ratio was computed as the ratio between the smaller cell volume and the mother cell volume. Three-dimensional triangular meshes of segmented cells and of their interfaces with neighbour cells were computed under AvizoFire (2013 Visualization Sciences Group, an FEI Company). The cell interface meshes were processed by a python script to automatically measure cell lengths along different directions. To this end, we first computed the intersection lines between side meshes by determining their shared vertices. Then, vertices at intersections

between three connected intersection lines were identified as cell corners. Cell lengths were obtained as Euclidean distances between corner vertices.

The number of faces per cell was computed using cell lineage trees with a python script. Mother cells were reconstructed up to the first embryonic cell by recursively merging sister cells. During this process, the generation at which each division plane had been formed was recorded. This allowed to determine for each observed cell the number of different generations at which interfaces with neighboring cells had been created. This number was taken as the number of faces for the cell. For the first embryonic cell, there were two interfaces, one corresponding to the wall separating this cell from the suspensor and the other corresponding to the separation with the outside of the embryo. At any generation $g$, the entropy of the distribution of cells among the different classes of cell shapes (defined by the number of faces) was computed as:

$$\text{Entropy}(g) = - \sum_f p_f(g) \log p_f(g)$$

where $p_f(g)$ designates the proportion of cells having $f$ faces at generation $g$. Entropy is a measure of the heterogeneity in a distribution: it is maximized for a uniform distribution; on the opposite, it takes its minimal value 0 when all individuals belong to the same class.

## Computer modeling of cell divisions

### Computer simulations

Cell divisions in reconstructed mother cells were simulated using the model we introduced previously (**Moukhtar et al., 2019**). This model takes as input the 3D binary mask of a mother cell and generates stochastically a partitioning of the cell based on geometric constraints, ignoring the environment of the cell within the tissue. For each simulation, the volume-ratio $\rho$ of the division (volume of the smaller daughter cell to the volume of the mother) was randomly drawn between 0.2 and 0.5. Each voxel of the mother cell mask was initially assigned to one or another of the two daughter cells with probability $\rho$ and $1 - \rho$, respectively. The Metropolis algorithm (**Metropolis et al., 1953**) was then used to iteratively minimize the interface area between the two daughter cells. The algorithm iterated Monte Carlo cycles of $N$ steps each, $N$ being the number of voxels in the binary mask of the mother cell. At each step, a voxel was randomly chosen. Its assignment to one or the other of the two daughter cells was flipped if this induced a decrease in the interface area. Otherwise, the flip was accepted with probability $\exp(-\beta \Delta A)$, where $\Delta A$ represented the change in interface area induced by the flip. The parameter $\beta$ was automatically adjusted at the end of each cycle so that about 5%, on average, of the candidate flips that would increase interface area were accepted. For each mother cell, 1000 independent simulations were run. The number of Monte Carlo cycles was set to 500, which was sufficient to ensure convergence (**Figure 6—figure supplement 1**).

### Scoring simulated divisions

The similarity between the simulated and observed divisions was scored based on their spatial overlap (**Figure 5—figure supplement 1**). Let $A$ and $B$ denote the sets of voxels in the two daughter cells of an observed division, and let $A'$ and $B'$ denote the two sets in a simulated division. The score quantifying the match between the two partitions of the mother cell space was defined as:

$$\text{score} = \max \left\{ \frac{|A \cap A'| + |B \cap B'|}{|A \cup B|}, \frac{|A \cap B'| + |B \cap A'|}{|A \cup B|} \right\}$$

This score varied between 0.5 (the minimum possible overlap) and 1.0 (perfect overlap).

## Acknowledgements

This work was supported by the MIA and BAP departments of INRA (funding support to EL) and has benefited from the support of IJPB's Plant Observatory technological platforms. We thank Herman Höfte for his comments and feedback on a first version of our manuscript. The IJPB benefits from the support of Saclay Plant Sciences-SPS (ANR-17-EUR-0007).

# Additional information

### Funding

| Funder | Grant reference number | Author |
|---|---|---|
| INRA MIA and INRA BAP departments | PhD Fellowship | Elise Laruelle |

The funders had no role in study design, data collection and interpretation, or the decision to submit the work for publication.

### Author contributions

Elise Laruelle, Conceptualization, Data curation, Software, Formal analysis, Validation, Investigation, Visualization, Methodology, Writing - original draft, Writing – review and editing; Katia Belcram, Resources, Data curation, Investigation; Alain Trubuil, Conceptualization, Resources, Supervision, Funding acquisition, Validation, Project administration, Writing – review and editing; Jean-Christophe Palauqui, Conceptualization, Supervision, Funding acquisition, Validation, Project administration, Writing – review and editing; Philippe Andrey, Conceptualization, Data curation, Software, Formal analysis, Supervision, Validation, Investigation, Visualization, Methodology, Writing - original draft, Project administration, Writing – review and editing

### Author ORCIDs

Elise Laruelle  http://orcid.org/0000-0003-3837-8183
Alain Trubuil  http://orcid.org/0000-0002-7861-0437
Jean-Christophe Palauqui  http://orcid.org/0000-0002-8816-4653
Philippe Andrey  http://orcid.org/0000-0001-5932-6863

### Decision letter and Author response

Decision letter https://doi.org/10.7554/eLife.79224.sa1
Author response https://doi.org/10.7554/eLife.79224.sa2

# Additional files

### Supplementary files
• MDAR checklist

### Data availability

The dataset of embryo images used in this study has been deposited in Data INRAE: Belcram, Katia; Palauqui, Jean-Christophe, 2022, "A collection of 3D images of *Arabidopsis thaliana* embryos", https://doi.org/10.15454/HIIBKW, Portail Data INRAE, V1. The TreeJ plugin for ImageJ/Fiji used for lineage reconstruction is available from https://imagej.net/plugins/treej and its source code from https://github.com/L-EL/TreeJ, (copy archived at swh:1:rev:5e70bb8149b9e18ba2439e24f4c-558cf649da348). The scripts used for cell shape analysis can be found at https://github.com/L-EL/plantCellShapeAnalysis, (copy archived at swh:1:rev:b681e0760416bee82d2dde15da96974602a9ff7c). The code source of the 3D cell division model, together with an executable version for Linux Ubuntu 20.04, has been deposited on Data INRAE: Philippe Andrey, 2022, Cell division model: source code and executable, https://doi.org/10.15454/LHGT6C, Portail Data INRAE, V1. Source Data files have been provided for Figures 2, 4, and 8.

The following dataset was generated:

| Author(s) | Year | Dataset title | Dataset URL | Database and Identifier |
|---|---|---|---|---|
| Balcram K, Palauqui JC | 2022 | A collection of 3D images of *Arabidopsis thaliana* embryos | https://doi.org/10.15454/HIIBKW | Data INRAE, 10.15454/HIIBKW |

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

## Appendix 1

### Predicting the topology of random divisions using graph cuts on polyhedral graphs

We consider the main three cell shapes observed during late embryogenesis in *Arabidopsis thaliana*. These shapes are the tetrahedron, triangular prism, and cuboid (containing 4, 5, and 6 faces, 4, 6, and 8 vertices, and 6, 9, and 12 edges, respectively). Our objective here is to enumerate the different ways of dividing these cell shapes and to characterize the resulting daughter shapes. The key to our analysis is to represent cell shapes as planar polyhedral graphs and cell divisions as graph cuts on these polyhedral graphs.

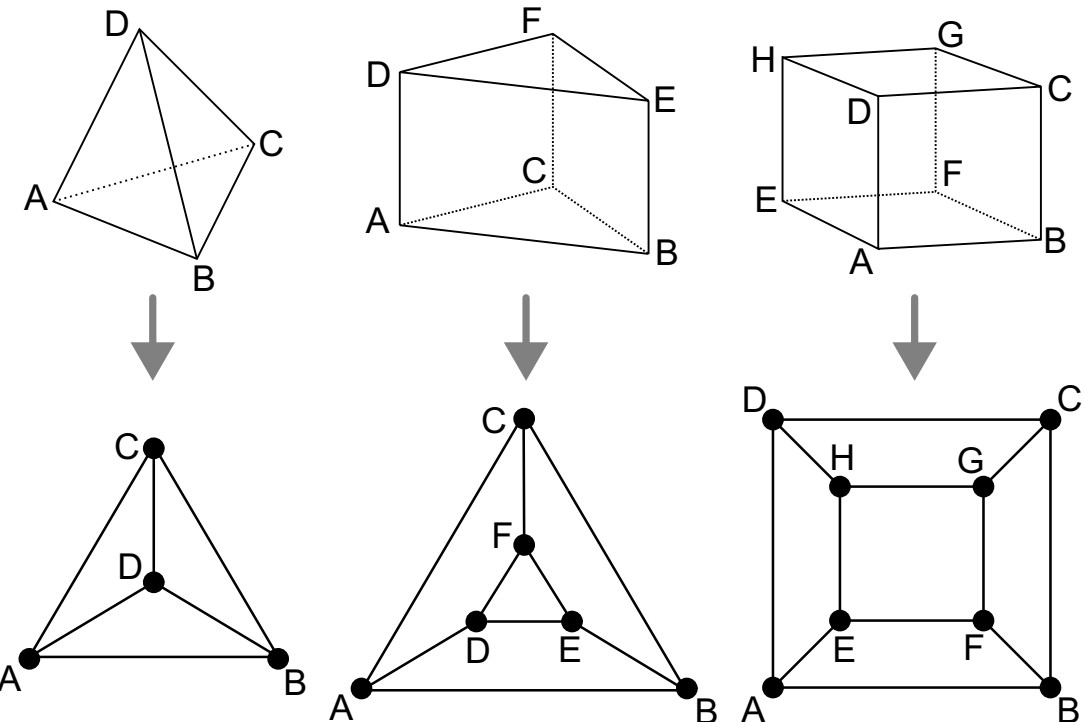

**Appendix 1—figure 1.** Polyhedral graphs (*Bottom row*) for the three main cell shapes (*Top row*) found in *Arabidopsis thaliana* early embryogenesis. Note that in these representations (Schlegel diagrams), the outside counts as one face of the corresponding polyhedron.

Any convex polyhedral cell shape with $F$ faces can be represented by a 3-connected planar graph $G$ of $V$ vertices inter-connected by $E$ edges (polyhedral graph). Such graphs can be represented in 2D by Schlegel diagrams (*Figure 1*). Because of the 3-connectivity of the corresponding graph, applying Euler's formula ($V - E + F = 2$) to any of these shapes gives the following relations:

$$2E = 3V$$

$$2F = 4 + V$$

Hence, we only need to determine the number of vertices of the daughter cells to characterize a cell division in terms of the abstract resulting cell shapes.

Given that cell divisions avoid existing vertices (avoidance of four-way junctions), any division splits the cell vertices in two disjoint sets of vertices. These sets are non-empty because cell division planes extend from one face of the mother cell to another one. Imposing that each face of the original mother cell is cut at most once implies that we do not consider curved division planes that would fold back to connect to the face from which they emanate. This is consistent with biological observations, given that situations where a division plane extends from an existing cell face to the same face are extremely rare in general and unknown in the embryo. Hence, a cell division corresponds to a graph cut, whereby a number of edges are removed to yield two disconnected subgraphs. Following cut, each subgraph is completed by adding new vertices at the cut positions.

A new edge is also introduced for each pair of new vertices located on the same face of the mother cell. The two resulting graphs are the graphs of the two daughter cells (*Figure 2*).

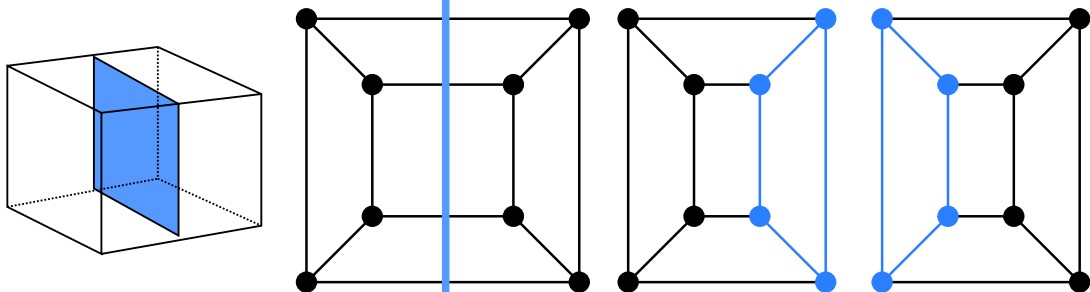

**Appendix 1—figure 2.** Cell division as cuts on polyhedral graphs: illustration with the division of a cuboid. The division on the left corresponds to the edge cut shown in the middle. Completing the two subgraphs of this cut with nodes and edges (*Blue*) yields the two subgraphs of the daughter cells. The obtained subgraphs correspond to two cuboids, as expected for the considered division.

A division can be characterized by a couple of integers $(p, q)$, where $p$ and $q$ are the number of mother cell vertices that are separated by the division. Since $q = V - p$, the division is actually fully characterized by $p$ only. We call $p$-division a division that separates $p$ vertices from the $V - p$ other vertices ($p > 0$). For example, the 1-divisions are the divisions whereby one of the vertices is separated from all the other ones ("corner" division). Since the $p$-divisions and the $q$-divisions with $q = V - p$ are two identical sets of divisions, we limit ourselves to situations where $p \leq q$, i.e., $p \leq V/2$.

We note $N(p)$ the number of possible $p$-divisions of a given cell shape. For each of these divisions, we note $K(p)$ the number of removed edges (=size of the edge cut-set); $V_p(p)$, $E_p(p)$ and $F_p(p)$ the total number of vertices, edges and faces in the daughter cell that inherits the $p$ vertices; $V_q(p)$, $E_q(p)$ and $F_q(p)$ the total number of vertices, edges and faces in the daughter cell that inherits the remaining $q = V - p$ vertices; $E_p^*(p)$ and $E_q^*(p)$ the number of edges that are inherited from the mother cell by each of these two daughter cells, respectively (=number of edges in the subgraphs of $G$ induced by the $p$ and $q$ vertices, respectively). We derive below the expressions of all these quantities as functions of $p$.

Each edge cut creates a new vertex for each daughter cell. We thus have, for any $p$:

$$V_p(p) = p + K(p)$$
$$V_q(p) = q + K(p)$$

Since each vertex is connected to three edges, the maximal number of possible cuts is $3p$ (remember that $p \leq q$). Each edge inherited by a daughter cell from its mother removes two potential cuts (one for each end-vertex). This gives:

$$K(p) = 3p - 2E_p^*(p)$$
$$= 3q - 2E_q^*(p)$$

We thus have:

$$V_p(p) = 2\left[2p - E_p^*(p)\right]$$
$$V_q(p) = 2\left[2q - E_q^*(p)\right]$$

which we can write

$$V_p(p) = 2\,Q_p(p)$$
$$V_q(p) = 2\,Q_q(p)$$

where

$$Q_p(p) = 2p - E_p^*(p)$$
$$Q_q(p) = 2q - E_q^*(p)$$

This finally leads to the following simple expressions for the number of edges and faces in the daughter cells:

$$E_p(p) = 3 Q_p(p)$$
$$E_q(p) = 3 Q_q(p)$$

$$F_p(p) = 2 + Q_p(p)$$
$$F_q(p) = 2 + Q_q(p)$$

Given that

$$E = E_p^*(p) + E_q^*(p) + K(p)$$

we also have

$$Q_q(p) = V/2 + p - E_p^*(p)$$

In a graph-theoretical perspective, we can thus fully describe a $p$-division and the resulting daughter cell shapes by two parameters only: the number $p$ of original vertices and the number $E_p^*(p)$ of original edges that are inherited by the "smallest" ($p \leq q$) of the two daughter cells.

To go further we must distinguish two situations, depending on whether the subgraph induced by the $p$ vertices and their $E_p^*(p)$ edges is cyclic or not.

If the subgraph induced by the $p$ vertices and their $E_p^*(p)$ edges contains no cycle (this is systematically the case for $p < 3$), then we have:

$$E_p^*(p) = p - 1$$

This gives the following features for a division induced by two acyclic subgraphs:

$$Q_p(p) = p + 1$$
$$Q_q(p) = V/2 + 1$$
$$V_p(p) = 2(p + 1)$$
$$V_q(p) = V + 2$$
$$E_p(p) = 3(p + 1)$$
$$E_q(p) = 3V/2 + 3$$
$$F_p(p) = p + 3$$
$$F_q(p) = V/2 + 3$$

One corollary of these results is that:

$$F_p(p) \leq F + 1$$
$$F_q(p) = F + 1$$

Hence, a division "in the acyclic case" systematically yields a daughter cell with one additional face compared with the mother. The other daughter cell has at most one additional face.

For the shapes we consider, we have $p \leq 4$. In this particular case, the presence of a cycle in the subgraph induced by the $p$ vertices ($p \geq 3$) and their $E_p^*(p)$ edges necessarily leads to:

$E_p^*(p) = p$ which yields the following features for a division induced by a cyclic subgraph:

$$Q_p(p) = p$$
$$Q_q(p) = V/2$$
$$V_p(p) = 2p$$
$$V_q(p) = V$$
$$E_p(p) = 3p$$
$$E_q(p) = 3V/2$$
$$F_p(p) = p + 2$$
$$F_q(p) = V/2 + 2$$

with, as a corollary, the following:

$$F_p \leq F$$
$$F_q = F$$

Hence, a division in the "cyclic case" cannot generate shapes with a larger number of faces than the mother cell. In addition, one of the two daughter cells has systematically the same shape as the mother cell.

Now it remains to enumerate the number $N(p)$ of different $p$-divisions for a given mother cell shape. The number of 1-divisions is simply:

$$N(1) = V$$

For the 2-divisions, we must distinguish the tetrahedral shape from the other ones because of symmetries of the 2-divisions in this shape:

$$N(2) = \begin{cases} E/2 & \text{if } V = 4 \\ E & \text{otherwise} \end{cases}$$

The number of 3-divisions (meaningful only for the two shapes with $V \geq 6$) is the number of pairs of adjacent edges in the mother cell graph. There are three pairs of adjacent edges at each vertex. For the triangular prismatic shape, care must be taken that the two triangular faces induce symmetries. On each face, there are indeed three pairs of edges that define the same division ("cyclic" case). Hence we have

$$N(3) = \begin{cases} 3V - 5 & \text{if } V = 6 \\ 3V & \text{if } V = 8 \end{cases}$$

The 4-divisions are meaningful only for the cuboidal shape. They are obtained either by separating opposite quadrilateral faces of the mother cell ("cyclic" case) or by separating one vertex and its three connected neighbors from the other four vertices ("acyclic" case). Taking care of symmetries, we thus have:

$$N(4) = F/2 + V/2$$
$$= 1 + \tfrac{3}{4}V$$

Now we can compute the expected proportions of cell shapes resulting from the division of a given cell shape, under a discrete uniform probability distribution over the space of possible divisions. In the sequel, we refer to each shape by the triplet $V.E.F$.

The possible outcomes of the division of a tetrahedral (4.6.4) mother cell are given in **Appendix 1— table 1**. From this table, we obtain that the expected proportions of cell shapes following the division of a 4.6.4 cell are:

$$\text{Daughters of 4.6.4} \begin{cases} P(4.6.4) = \frac{4}{14} & (28.6\%) \\ P(6.9.5) = \frac{10}{14} & (71.4\%) \end{cases}$$

**Appendix 1—table 1.** Divisions of the tetrahedral cell shape (4.6.4).

| $p/V - p$ | 1/3 | 2/2 |
|---|---|---|
| $N(p)$ | 4 | 3 |
| $p$-shape | 4.6.4 | 6.9.5 |
| $q$-shape | 6.9.5 | 6.9.5 |

The possible outcomes of the division of a triangular prismatic (6.9.5) mother cell are given in **Appendix 1—table 2**. From this table, we obtain that the expected proportions of cell shapes following the division of a 6.9.5 cell are:

$$\text{Daughters of 6.9.5} \begin{cases} P(4.6.4) & = \frac{6}{56} & (10.7\%) \\ P(6.9.5) & = \frac{11}{56} & (19.6\%) \\ P(8.12.6) & = \frac{39}{56} & (69.9\%) \end{cases}$$

**Appendix 1—table 2.** Divisions of the triangular prismatic cell shape (6.9.5).
$\alpha$ refers to the case where the $p$-subgraph is acyclic, $\beta$ to the case where it is cyclic.

| $p/V - p$ | 1/5 | 2/4 | 3/3α | 3/3β |
|---|---|---|---|---|
| $N(p)$ | 6 | 9 | 12 | 1 |
| $p$-shape | 4.6.4 | 6.9.5 | 8.12.6 | 6.9.5 |
| $q$-shape | 8.12.6 | 8.12.6 | 8.12.6 | 6.9.5 |

The possible outcomes of the division of a cuboidal (8.12.6) mother cell are given in **Appendix 1—table 3**. From this table, we obtain that the expected proportions of cell shapes following the division of a 8.12.6 cell are:

**Appendix 1—table 3.** Divisions of the cuboidal cell shape (8.12.6).
$\alpha$ refers to the case where the $p$-subgraph is acyclic, $\beta$ to the case where it is cyclic.

| $p/V - p$ | 1/7 | 2/6 | 3/5 | 4/4α | 4/4β |
|---|---|---|---|---|---|
| $N(p)$ | 8 | 12 | 18 | 4 | 3 |
| $p$-shape | 4.6.4 | 6.9.5 | 8.12.6 | 10.15.7 | 8.12.6 |
| $q$-shape | 10.15.7 | 10.15.7 | 10.15.7 | 10.15.7 | 8.12.6 |

$$\text{Daughters of 8.12.6} \begin{cases} P(4.6.4) & = \frac{8}{90} & (8.9\%) \\ P(6.9.5) & = \frac{12}{90} & (13.3\%) \\ P(8.12.6) & = \frac{24}{90} & (26.7\%) \\ P(10.15.7) & = \frac{46}{90} & (51.1\%) \end{cases}$$

