## [Editor Report]

This manuscript presents a new and interesting work exploring stochastic and deterministic aspects of embryonic cell division in plants. In particular, the power of the proposed approach lies in the quantitative analysis of 3D cell geometries that is combined with quantitative computer modelling.

---

## [Decision Letter]

**Decision letter after peer review:**

Thank you for submitting your article "Large-scale analysis and computer modeling reveal hidden regularities behind variability of cell division patterns in *Arabidopsis thaliana* embryogenesis" for consideration by *eLife*. Your article has been reviewed by 3 peer reviewers, one of whom is a member of our Board of Reviewing Editors, and the evaluation has been overseen by Jürgen Kleine-Vehn as the Senior Editor. The reviewers have opted to remain anonymous.

Essential revisions:

1) Attractor patterns seem descriptive and the conclusions seem to be based on qualitative observation of a few cases. Could authors provide more quantitative measurements and or simulations to strengthen their claims?

2) There is no data to support the claim of the abstract that 'variability in cell division patterns of Arabidopsis embryos is accompanied by a progressive reduction of cell shape heterogeneity'. Shall authors provide the essential data or alternatively modify the claims.

3) The manuscript should improve explanations that would help non-mathematicians to understand the concepts behind the methods and tools applied.

4) Some parts of the manuscript seem purely descriptive. What is the meaning of observed patterns for general plant development and patterning?

5) Models would benefit from sensitivity analysis i.e. how the choice of vertices affects model robustness (see also specific comments from three reviewers).

Please find specific comments of three Reviewers below:

*Reviewer #1 (Recommendations for the authors):*

The manuscript would benefit from a more elaborate description of the relevance of observed patterns for embryogenesis and additional analysis combined with a detailed explanation of the computer models.

*Reviewer #2 (Recommendations for the authors):*

1) I couldn't find any data in the manuscript supporting the claim of the abstract that 'variability in cell division patterns of Arabidopsis embryos is accompanied by a progressive reduction of cell shape heterogeneity'

2) Last paragraph page 13, it would be helpful to state explicitly that the 3 possible division planes are 'statistically' equivalent.

3) An additional discussion point could be that the rules proposed based on the division of epidermal cells in 2D could be probabilistic because of the 'hidden' 3D imperfect geometry of cells.

*Reviewer #3 (Recommendations for the authors):*

Overall, this manuscript provides important results and hypotheses about the mechanism relating to cellular patterning in plant development. However, as I mentioned in the Public review part, this manuscript lacks explanations that are essential to understand the context of the manuscript properly, especially for non-mathematical background. This problem lost my confidence to correctly evaluate the novelty and superiority. Therefore, the authors need to describe in the main body the detail of some points as follows even if they are mentioned in the supplemental part.

1) Relating to the reconstruction of cellular lineage, I couldn't follow how authors can determine the correct cell pairs derived from the same mother cell without time-lapse observation backing to the 1C stage. The authors described "Note that the absence of embryo bending at these stages ensured that the plane orientation in the embryo at the time of division could be correctly inferred even for patterns reconstructed from later stages", but, for example, if four cells are positioned in a crossed arrangement, some criterion should be needed to estimate whether longitudinal or transverse division occurred first.

2) How do the criteria in Figure 2B contribute to subsequent analyses?

3) In Figure 3, it's difficult to understand how the quantification of cellular lineage was achieved because I failed to understand the exact meaning of the sentence "Frequencies were computed…..back to stage 16C stage from observed configuration." in the caption.

4) It's a little hard to find what Figures 4A and B are showing.

5) In Figures 5-7, it's hard to understand the conclusion because it is unclear which points in the left panels are corresponding to each division pattern in the right panels.

6) In Figure 7, how can we conclude that the variability is buffered in the latter stages without time-lapse observation of the entire process of the single embryo?

---

## [Author Response]

Essential revisions:Reviewer #1 (Recommendations for the authors):The manuscript would benefit from a more elaborate description of the relevance of observed patterns for embryogenesis and additional analysis combined with a detailed explanation of the computer models.

We believe the modifications we made when answering the Public Review recommendations made by the Reviewer above should address this general recommendation.

Reviewer #2 (Recommendations for the authors):1) I couldn't find any data in the manuscript supporting the claim of the abstract that 'variability in cell division patterns of Arabidopsis embryos is accompanied by a progressive reduction of cell shape heterogeneity'

The data were presented in Figures 2 and 3.

First, Figure 2E displays the proportion of the three main categories of cell shapes at each generation between G1 and G8. It illustrates that cell shape heterogeneity is maximal, at the embryo scale, at stage 16C, with a distribution over tetrahedron (25%), triangular prism (50%), and cuboid (25%). From stage 16C, reduction in cell shape heterogeneity results from the increasing proportion of the cuboid shape, which progressively takes over the two others, representing more than 90% at G8. This reveals a strong homogenization of cell shapes. Second, variability in cell division patterns is illustrated in Figure 3 and its Figure Supplements 1 and 2 (Supplementary Figures S3 and S4 in the initial submission), which display cell lineage trees in the four embryo domains from stage 16C, with frequencies observed among different cell division patterns. These trees show variable cell division patterns and the existence of multiple division pathways from G4 to G8.

Altogether, we thus believe our original manuscript already provided the quantitative data supporting the claim that the variability that appears at G4 in cell division patterns is paralleled by a reduction in the variability of cell shapes. To make our claim stronger, we quantified cell shape heterogeneity based on the entropy of the distribution of the number of faces at each generation (definition provided in Material and Methods section). The plot of entropy dynamics we added as an overlay on Figure 2E corroborates our claim that variability in cell shapes increases until G4 (when division patterns are stereotyped) and continuously decreases beyond this generation (when division patterns become variable).

2) Last paragraph page 13, it would be helpful to state explicitly that the 3 possible division planes are 'statistically' equivalent.

As already stated in the Discussion, the three orientations are statistically equivalent only in the case of a perfect invariance by rotation, as would be the case if cell shapes were triangular prisms with an equilateral base. Departure from perfect shape symmetry induces asymmetry between the three local minima, one of which becomes a global optimum. The modification made to Figures 6 and 7 according to Reviewer 3’s suggestion (showing in the dot plots the positions of the simulated patterns) makes this clearer in the revision. For clarification, we have also reformulated the conclusion of the last paragraph on page 13 in the original version (now on page 12). We have also added a new Figure 10 to illustrate this interpretation in the Discussion.

3) An additional discussion point could be that the rules proposed based on the division of epidermal cells in 2D could be probabilistic because of the 'hidden' 3D imperfect geometry of cells.

We thank the Reviewer for this very interesting suggestion. Completely addressing this issue would require comparing analyses in 2D and in 3D of the same division patterns in the same tissues. Previous studies on stochastic cell division rules in plants (Besson and Dumais 2011) analyzed other systems than the *Arabidopsis* embryo. In addition, many division patterns in the embryo cannot be reduced to 2D representations. Hence, we fear it would be too speculative at this stage to develop this point in the present manuscript.

Reviewer #3 (Recommendations for the authors):Overall, this manuscript provides important results and hypotheses about the mechanism relating to cellular patterning in plant development. However, as I mentioned in the Public review part, this manuscript lacks explanations that are essential to understand the context of the manuscript properly, especially for non-mathematical background. This problem lost my confidence to correctly evaluate the novelty and superiority. Therefore, the authors need to describe in the main body the detail of some points as follows even if they are mentioned in the supplemental part.1) Relating to the reconstruction of cellular lineage, I couldn't follow how authors can determine the correct cell pairs derived from the same mother cell without time-lapse observation backing to the 1C stage. The authors described "Note that the absence of embryo bending at these stages ensured that the plane orientation in the embryo at the time of division could be correctly inferred even for patterns reconstructed from later stages", but, for example, if four cells are positioned in a crossed arrangement, some criterion should be needed to estimate whether longitudinal or transverse division occurred first.

The absence of bending or rotation of the embryo is important to correctly infer the relative orientation of the division plane within the coordinate frame of the embryo at the time of division from a later stage. This is independent from the question of determining sister cells and reconstructing cell lineages. For this purpose, we exploited the fact that actual four-way junctions are rare in 3D. At the 2C-4C transition and at later stages in the outer basal domain, for example, there were frequent patterns of four cells forming a crossed arrangement at their external surface. However, examining the cell junctions in 3D most frequently revealed a transition from a cross pattern to a ‘double-T’ pattern when penetrating inside the tissue, thus revealing a former division plane that had been reached on opposite sides by two more recent ones. Overall, we observed 28/4285 (0.7%) division patterns where ambiguity could not be resolved in 3D. Roughly half of these patterns were observed at the 2C-4C generation of divisions and had no impact on our subsequent analyses because of the symmetry between the two possible interpretations. The other patterns were mostly located in the outer domain at advanced stages. These few cases were assigned to the most frequent configuration observed over the whole dataset.

We have added these precisions to the Material and Methods section and have also reported in the main text the principle on which lineage reconstruction was based.

2) How do the criteria in Figure 2B contribute to subsequent analyses?

Figure 2B illustrates how the proposed new topological descriptor of cell shape (number of cell faces) is computed. This is the basis for the cell shape classification reported in Figure 2C-I and for cell-shape based analyses (first four sub-sections of the Results sections). As explained in Section “Diversity in cell shape is domain-specific” (third paragraph), this descriptor corresponds to the number of planes that contributed, since the 1C stage, to generate a given cell shape. Key with this descriptor is to depend on the relative orientation of division within the mother cell (i.e., on the mother cell faces that are intersected by a division plane), but not on the precise positioning, for a given orientation, of the plane. By separating topology from geometry, using this descriptor allows relating variability in division plane orientation to the resulting cell shapes. A key result of our analysis based on this descriptor is to reveal that strong constraints are exerted on the orientation of division planes (Figure 4C) in spite of observed variability in division patterns (Figure 1).

In an attempt to clarify the importance of distinguishing between cell shape and cell topology, we have reformulated and restructured the introductory text of Section 2.1/“Diversity in cell shape is domain-specific”.

3) In Figure 3, it's difficult to understand how the quantification of cellular lineage was achieved because I failed to understand the exact meaning of the sentence "Frequencies were computed…..back to stage 16C stage from observed configuration." in the caption.

At any generation, the reported frequencies were computed using embryos that were observed at this generation and also using embryos that were observed at later generations. For the latter, we used their cell lineages to recursively merge sister cells until reaching the considered generation. We have reformulated the incriminated sentence of the caption to clarify.

4) It's a little hard to find what Figures 4A and B are showing.

The top row in Figure 4A shows the three main cell shapes observed in the embryo up to Generation 8 (same as Figure 2D). These shapes are polyhedra composed of vertices, of edges joining these vertices, and of faces delineated by edges. The bottom row of Figure 4A shows the corresponding polyhedral graphs, or Schlegel diagrams. These 2D representations are obtained by virtually projecting (flattening) the 3D cell shapes in a direction orthogonal to one of their faces. Schlegel diagrams have the same number of vertices, edges, and faces as their corresponding polyhedra (the outside of a diagram represents the face used to perform the projection).

Figure 4B illustrates how one can interpret a cell division as a cut in a polyhedral graph, thus generating two new graphs that correspond to the shapes of the daughter cells. By enumerating all possible cuts for a given mother cell graph, one can compute the theoretical distributions shown in Figure 4C.

This graph-theoretical approach allowed us to compute the expected outcome of cell divisions if cell division was topologically random (i.e., not taking into account edge lengths). The strong discrepancy between observed and predicted distributions shows that topology alone was not sufficient to predict division orientations, confirming that additional factors control the orientation of divisions.

We have slightly redesigned Figure 4 and completed its caption to make the correspondence between shapes and graphs clearer; we have also enriched the corresponding figure in the Appendix to clarify the passage from a 3D polyhedron to its corresponding 2D graph (AppendixFigure 1). Lastly, we have reformulated and clarified the text in the first paragraph of Section 2.4/”Graph theory of cell division reveals variability is constrained”.

5) In Figures 5-7, it's hard to understand the conclusion because it is unclear which points in the left panels are corresponding to each division pattern in the right panels.

Figure 5 already contained numbers showing the correspondence between the different division patterns and their positions in the dot plot. We have added colored squares in the dot plots of Figures 6 and 7 to show the locations in the dot plots of the displayed simulated patterns. In all cases but the first and last ones in Figure 7, the simulated pattern matching best with observation is found at the bottom left of the dot plot, supporting the conclusion that observed patterns correspond to the geometrical rule (area minimization conditioned on passing through the cell center).

6) In Figure 7, how can we conclude that the variability is buffered in the latter stages without time-lapse observation of the entire process of the single embryo?

The Reviewer is probably referring to Figure 9 rather than Figure 7, which is not related to attractor patterns. This question is also related to Comment 1 by the Reviewer about the construction of cell lineages. The key point is that, as detailed in our answer to Comment 1 above, the cell patterns in an embryo observed at a given stage bear information about the history of successive divisions that occurred at previous stages. It is thus possible to identify that different sequences of division orientations led to similar patterns, as illustrated in different domains in Figure 9. Hence, time-lapse is not required because we do not predict the future evolution of the embryo from a given stage but instead reconstruct its past history from its observed stage.